# IW-GAE: Importance weighted group accuracy estimation for improved calibration and model selection in unsupervised domain adaptation

## Abstract

Reasoning about a model's accuracy on a test sample from its confidence is a central problem in machine learning, being connected to important applications such as uncertainty representation, model selection, and exploration. While these connections have been well-studied in the i.i.d. settings, distribution shifts pose significant challenges to the traditional methods. Therefore, model calibration and model selection remain challenging in the unsupervised domain adaptation problem–a scenario where the goal is to perform well in a distribution shifted domain without labels. In this work, we tackle difficulties coming from distribution shifts by developing a novel importance weighted group accuracy estimator. Specifically, we formulate an optimization problem for finding an importance weight that leads to an accurate group accuracy estimation in the distribution shifted domain with theoretical analyses. Extensive experiments show the effectiveness of group accuracy estimation on model calibration and model selection. Our results emphasize the significance of group accuracy estimation for addressing challenges in unsupervised domain adaptation, as an orthogonal improvement direction with improving transferability of accuracy.

## 1 Introduction

In this work, we consider a classification problem in unsupervised domain adaptation (UDA). UDA aims to transfer knowledge from a source domain with ample labeled data to enhance the performance in a target domain where labeled data is unavailable. In UDA, the source and target domains have *different data generating distributions*, so the core challenge is to transfer knowledge contained in the labeled dataset in the source domain to the target domain under the distribution shifts. Over the decades, significant improvements in the transferability from source to target domains have been made, resulting in areas like domain alignment (Ben-David et al., 2010; Ganin et al., 2016; Long et al., 2018; Zhang et al., 2019) and self-training (Chen et al., 2020; Cai et al., 2021; Liu et al., 2021).

Improving calibration performance, which is about matching predictions regarding a random event to the long-term occurrence of the event (Dawid, 1982), is of central interest in the machine learning community due to its significance to safe and trustworthy deployment of machine learning models in critical real-world decision-making systems (Lee and See, 2004; Amodei et al., 2016). In independent and identically distributed (i.i.d.) settings, calibration performance has been significantly improved by various approaches (Guo et al., 2017; Gal and Ghahramani, 2016; Lakshminarayanan et al., 2017). However, producing well-calibrated predictions in UDA remains challenging due to the distribution shifts. Specifically, Wang et al. (2020) show the discernible compromise in calibration performance as an offset against the enhancement of target accuracy. A further observation reveals that state-of-the-art calibrated classifiers in the i.i.d. settings begin to generate unreliable uncertainty representation in the face of distributional shifts (Ovadia et al., 2019). As such, enhancing the calibration performance in UDA requires carefully addressing the impacts of the distribution shifts.

Moreover, the model selection task in UDA remains challenging due to the scarcity of labeled target domain data that are required to evaluate model performance. In the i.i.d. settings, a standard approach for model selection is a cross-validation method—constructing a hold-out dataset for selecting the model that yields the best performance on the hold-out dataset. While cross-validation provides favorable statistical guarantees (Stone, 1977; Kohavi et al., 1995), such guarantees falter

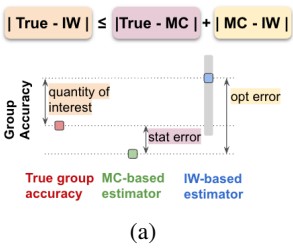 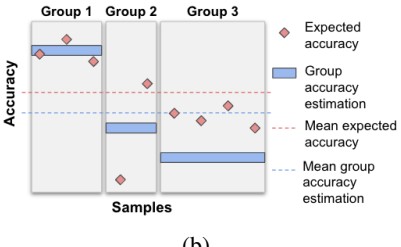

(a)  (b)

Figure 1: In Figure 1(a), a shaded area for the IW-based estimator represents possible estimations from IWs in the confidence interval. Figure 1(b) illustrates both ideal and failure cases of IW-GAE with nine data points (red diamonds) from three groups (gray boxes). Group 1 is desirable for model calibration where the group accuracy estimation (a blue rectangle) well represents the individual expected accuracies of samples in the group. Conversely, group accuracy estimation could inaccurately represent the individual accuracies in the group due to a high variance of accuracies within the group (group 2) and a high bias of the estimator (group 3).

in the presence of the distribution shifts due to the violation of the i.i.d. assumption. In practice, it has also been observed that performances of machine learning models measured in one domain have significant discrepancy to their performances in another distribution shifted domain (Hendrycks and Dieterich, 2019; Ovadia et al., 2019; Recht et al., 2019). Therefore, applying model selection techniques in the i.i.d. settings to the labeled source domain is suboptimal in the target domain.

This paper proposes **importance weighted group accuracy estimation (IW-GAE)** that *simultaneously addresses these critical aspects in UDA* from a new perspective of predicting a group accuracy. We partition predictions into a set of groups and then estimate the group accuracy–the average accuracy of predictions in a group–by importance weighting. When *the group accuracy estimate accurately represents the expected accuracy* of a model for individual samples in the group (e.g., group 1 in Figure 1(b)), using the group accuracy estimate as prediction confidence induces a well-calibrated classifier. When *the average of the group accuracy estimates matches the expected accuracy (e.g., two dotted lines in Figure 1(b) are close to each other)*, the average group accuracy becomes a good model selection criterion. In this work, we formulate a novel optimization problem for finding importance weights (IWs) that induce a group accuracy estimator satisfying these ideal properties under the distribution shifts. Specifically, we define two estimators for *the group accuracy in the source domain*, where only one of them depends on the IW. Then, we find the IW that makes the two estimators close to each other by solving the optimization problem (cf. reducing opt error in Figure 1(a)). Through a theoretical analysis and several experiments, we show that the optimization process results in *an accurate group accuracy estimator for the target domain* (cf. small quantity of interest in Figure 1(a)), thereby improving model calibration and model selection performances.

Our contributions can be summarized as follows: 1) We propose a novel optimization problem for IW estimation that can directly reduce an error of the quantity of interests in UDA with a theoretical analysis; 2) We show when and why considering group accuracy, instead of the accuracy for individual samples, is statistically favorable based on the bias-variance decomposition analysis, which can simultaneously benefit model calibration and model selection; 3) On average, IW-GAE improves state-of-the-art by 26% in the model calibration task and 14% in the model selection task.

## 2 RELATED WORK

**Model calibration in UDA** Although post-hoc calibration methods Guo et al. (2017) and Bayesian methods (Gal and Ghahramani, 2016; Lakshminarayanan et al., 2017; Sensoy et al., 2018) have been achieving impressive calibration performances in the i.i.d. setting, it has been shown that most of the calibration improvement methods fall short under distribution shifts (Ovadia et al., 2019) (see Appendix B.1 for more discussion). While handling model calibration problems under general distribution shifts is challenging, the availability of unlabelled samples in the distribution shifted target domain relaxes the difficulty in UDA. In particular, unlabeled samples in the target domain enable an IW formulation for the quantity of interests in the shifted domain. Therefore, the post-doc calibration methods (e.g., Guo et al. (2017)) can be applied by reweighting calibration measures such as the expected calibration error (Wang et al., 2020) and the Brier score (Park et al., 2020) in the source dataset with an IW. However, estimating the IW brings another difficulty of high-dimensional density estimation. In this work, instead of concentrating on obtaining accurate importance weighted

calibration measures for matching the maximum softmax output to the expected accuracy, we aim to directly estimate the accuracy in the distribution shifted target domain.

**Model selection in UDA** A standard procedure for model selection in the i.i.d. settings is the cross-validation, which enjoys statistical guarantees about bias and variance of model performance (Stone, 1977; Kohavi et al., 1995; Efron and Tibshirani, 1997). However, in UDA, the distribution shifts violate assumptions for the statistical guarantees. Furthermore, in practice, the accuracy measured in one domain is significantly changed in the face of natural/adversarial distribution shifts (Goodfellow et al., 2015; Hendrycks and Dietterich, 2019; Ovadia et al., 2019). To tackle the distribution shift problem, importance weighted cross validation (Sugiyama et al., 2007) applies importance sampling for obtaining an unbiased estimate of model performance in the distribution shifted target domain. Further, recent work in UDA controls variance of the importance-weighted cross validation with a control variate (You et al., 2019). These methods aim to accurately estimate the IW and then use an IW formula for the expected accuracy estimation. In this work, our method concerns the accuracy estimation error in the target domain during the process of IW estimation, which can potentially induce an IW estimation error but resulting in an accurate accuracy estimator.

## 3 BACKGROUND

**Notation and problem setup** Let $\mathcal{X} \subseteq \mathbb{R}^r$ and $\mathcal{Y} = [K] := \{1, 2, \cdots, K\}$ be input and label spaces. Let $\hat{Y} : \mathcal{X} \to [K]$ be the prediction function of a model and $Y(x)$ is a (conditional) $K$-dimensional categorical random variable related to a label at $X = x$. When there is no ambiguity, we represent $Y(x)$ and $\hat{Y}(x)$ as $Y$ and $\hat{Y}$ for brevity. We are given a labeled source dataset $\mathcal{D}_S = \{(x_i^{(S)}, y_i^{(S)})\}_{i=1}^{N^{(S)}}$ sampled from $p_{S_{XY}}$ and an unlabeled target dataset $\mathcal{D}_T = \{x_i^{(T)}\}_{i=1}^{N^{(T)}}$ sampled from $p_{T_X}$ where $p_{S_{XY}}$ is a joint data generating distribution of the source domain and $p_{T_X}$ is a marginal data generating distribution of the target domain. We also denote $\mathbb{E}_p[\cdot]$ as the population expectation and $\hat{\mathbb{E}}_p[\cdot]$ as its empirical counterpart. For $p_{S_{XY}}$ and $p_{T_{XY}}$, we consider a covariate shift without a concept shift; i.e., $p_{S_X}(x) \neq p_{T_X}(x)$ but $p_{S_{Y|X}}(y|x) = p_{T_{Y|X}}(y|x)$ for all $x \in \mathcal{X}$. For the rest of the paper, we use the same notation for marginal and joint distributions when there is no ambiguity; that is, $\mathbb{E}_{p_S}[u_1(X)] = \mathbb{E}_{p_{S_X}}[u_1(X)]$ and $\mathbb{E}_{p_S}[u_2(X,Y)] = \mathbb{E}_{p_{S_{XY}}}[u_2(X,Y)]$. However, we use the explicit notation for the conditional distribution as $p_{S_{Y|X}}$ and $p_{T_{Y|X}}$ to avoid confusion.

In this work, we consider an IW estimation problem for improving model calibration and model selection in UDA. Importance weighting can address many problems in UDA due to its statistical exactness for dealing with two different probability distributions under the absolute continuity condition (Horvitz and Thompson, 1952; Sugiyama et al., 2007) that is often assumed in the literature. Specifically, for densities $p_S$ and $p_T$, a quantity of interest $u(\cdot, \cdot)$ in $p_T$ can be computed by $\mathbb{E}_{p_T}[u(X,Y)] = \mathbb{E}_{p_S}[w^*(X)u(X,Y)]$ where $w^*(x) := \frac{p_T(x)}{p_S(x)}$ is the IW of $x$. We next review two main approaches for the IW estimation, which circumvent the challenges of directly estimating the IW, or the densities $p_S$ and $p_T$, in a high-dimensional space.

**Estimating IW by discriminative learning** Bickel et al. (2007) formulate the IW estimation into a discriminative learning problem by applying the Bayes' rule, which is more sample efficient (Ng and Jordan, 2001; Tu, 2007; Long and Servedio, 2006). Specifically, with a discriminative model that classifies source and target samples, the IW can be computed as $w^*(x) = \frac{p_T(x)}{p_S(x)} = \frac{\nu(x|d=1)}{\nu(x|d=0)} = \frac{P(d=0)}{P(d=1)} \frac{P(d=1|x)}{P(d=0|x)}$ where $\nu$ is a distribution over $(x, d) \in (\mathcal{X} \times \{0, 1\})$ and $d$ is a Bernoulli random variable indicating whether $x$ belongs to the target domain or not. For the IW estimation, $P(d=0)/P(d=1)$ can be estimated by counting sample sizes of $\mathcal{D}_S$ and $\mathcal{D}_T$. Also, to estimate $P(d=1|x)/P(d=0|x)$, a logistic regression can be trained by assigning a domain index of zero to $x_S \in \mathcal{D}_S$ and one to $x_T \in \mathcal{D}_T$, and maximizing log-likelihood with respect to domain datasets.

**Estimating confidence interval of importance weight** Recently, nonparametric estimation of the IW is proposed in the context of generating a probably approximately correct (PAC) prediction set (Park et al., 2022). In this approach, $\mathcal{X}$ is partitioned into $B$ number of bins ($\mathcal{X} = \cup_{i=1}^B \mathcal{B}_i$) with

$$I^{(B)} : \mathcal{X} \to [B] \text{ such that } \mathcal{B}_i = \{x \in \mathcal{X} | I^{(B)}(x) = i\}, \quad i \in [B]. \tag{1}$$

Under the partitions, the binned probabilities $\bar{p}_S(x) = \bar{p}_{S_{I^{(B)}(x)}}$ with $\bar{p}_{S_j} = \int_{\mathcal{B}_j} p_S(x)dx$ and $\bar{p}_T(x) = \bar{p}_{T_{I^{(B)}(x)}}$ with $\bar{p}_{T_j} = \int_{\mathcal{B}_j} p_T(x)dx$ are defined. Then, the confidence intervals (CIs) of the

IW in $\mathcal{B}_j$ can be obtained by applying the Clopper–Pearson CI (Clopper and Pearson, 1934) to the binned probabilities $p_{S_j}$ and $p_{T_j}$ for $j \in [B]$ (Park et al., 2022). Specifically, for $\bar{\delta} := \delta/2B$, the following inequality holds with probability at least $1 - \delta$:

$$\frac{[\underline{\theta}(n_j^{(T)};N^{(T)},\bar{\delta})-G]^+}{\bar{\theta}(n_j^{(S)};N^{(S)},\bar{\delta})+G} \leq w_j^* := \frac{\bar{p}_{T_j}}{\bar{p}_{S_j}} \leq \frac{\bar{\theta}(n_j^{(T)};N^{(T)},\bar{\delta})+G}{[\underline{\theta}(n_j^{(S)};N^{(S)},\bar{\delta})-G]^+} \tag{2}$$

where $\bar{\theta}(k;m,\delta) := \inf\{\theta \in [0,1] | F(k;m,\theta) \leq \delta\}$ and $\underline{\theta}(k;m,\delta) := \sup\{\theta \in [0,1]|F(k;m,\theta) \geq \delta\}$ with $F$ being the cumulative distribution function of the binomial distribution and $G \in \mathbb{R}_+$ is a constant that satisfies $\int_{\mathcal{B}_j} |p_S(x) - p_S(x')|dx' \leq G$ and $\int_{\mathcal{B}_j} |p_T(x) - p_T(x')|dx' \leq G$ for all $x \in \mathcal{B}_j$ and $j \in [B]$. For the rest of the paper, we refer to $\{w_i^*\}_{i \in B}$ as **binned IWs**. Also, we let $\Phi_j := \left[\frac{[\underline{\theta}(n_j^{(T)};N^{(T)},\bar{\delta})-G]^+}{\bar{\theta}(n_j^{(S)};N^{(S)},\bar{\delta})+G}, \frac{\bar{\theta}(n_j^{(T)};N^{(T)},\bar{\delta})+G}{[\underline{\theta}(n_j^{(S)};N^{(S)},\bar{\delta})-G]^+}\right]$ be the CI of $w_i^*$.

## 4 IMPORTANCE WEIGHTED GROUP ACCURACY ESTIMATION

In this section, we propose IW-GAE that estimates the group accuracy in the target domain for addressing model calibration and selection tasks in UDA. Specifically, we construct $M$ groups denoted by $\{\mathcal{G}_i\}_{i \in [M]}$ and then estimate the average accuracy of each group in the target domain with IW. To this end, we define the **target group accuracy** of a group $\mathcal{G}_n$ with the true IW $w^*$ as

$$\alpha_T(\mathcal{G}_n; w^*) := \mathbb{E}_{p_S}\left[w^*(X)\mathbf{1}(Y = \hat{Y})|X \in \mathcal{G}_n\right]\frac{P(X_S \in \mathcal{G}_n)}{P(X_T \in \mathcal{G}_n)} \tag{3}$$

where $X_S$ and $X_T$ are random variables having densities $p_S$ and $p_T$, respectively. It is called the group accuracy because $\mathbb{E}_{p_S}\left[w^*(X)\mathbf{1}(Y = \hat{Y})|X \in \mathcal{G}_n\right]\frac{P(X_S \in \mathcal{G}_n)}{P(X_T \in \mathcal{G}_n)} = \int_{x \in \mathcal{X}} \mathbf{1}(Y = \hat{Y})\frac{p_T(x)\mathbf{1}(x \in \mathcal{G}_n)}{P(X_T \in \mathcal{G}_n)}dx = \mathbb{E}_{p_T}\left[\mathbf{1}(Y(X) = \hat{Y}(X))|X \in \mathcal{G}_n\right]$. We denote $\hat{\alpha}_T(\mathcal{G}_n; w^*)$ to be the expectation with respect to the empirical measure. We also define the **source group accuracy** as

$$\alpha_S(\mathcal{G}_n; w^*) := \mathbb{E}_{p_T}\left[\frac{\mathbf{1}(Y(X)=\hat{Y}(X))}{w^*(X)}|X \in \mathcal{G}_n\right]\frac{P(X_T \in \mathcal{G}_n)}{P(X_S \in \mathcal{G}_n)}. \tag{4}$$

Once we obtain an IW estimation $\hat{w} : \mathcal{X} \to \mathbb{R}_+$ and a group assignment $I^{(g)} : \mathcal{X} \to [M]$ with methods described in Section 4.2, IW-GAE can estimate the group accuracy, denoted as $\hat{\alpha}_T(\mathcal{G}_i; \hat{w})$, that can be used to simultaneously solve model calibration and model selection tasks with attractive properties. Specifically, for model calibration, previous approaches (Park et al., 2020; Wang et al., 2020) depend on a temperature scaling method (Guo et al., 2017) that does not provide theoretical guarantees about the calibration error. In contrast, IW-GAE uses $\hat{\alpha}_T(\mathcal{G}_{I^{(g)}(x)}; \hat{w})$ as an estimate of confidence for a test sample $x \sim p_T$. Therefore, due to the guarantees about the group accuracy estimation error (cf. Proposition 4.2 and (5)), *IW-GAE enjoys a bounded calibration error*. For model selection, IW-GAE uses average group accuracy $\hat{\mathbb{E}}_{p_T}[\hat{\alpha}_T(\mathcal{G}_{I^{(g)}(X)}; \hat{w})]$ computed with $\mathcal{D}_T$ as a model selection criterion. While the previous approaches (Sugiyama et al., 2007; You et al., 2019) also aim to estimate the model accuracy in the target domain, *IW-GAE considers an additional regularization encouraging accurate group accuracy estimation for each group*.

### 4.1 MOTIVATION FOR ESTIMATING THE GROUP ACCURACY

First, we motivate the idea of predicting group accuracy, instead of an expected accuracy for each sample. Suppose we are given samples $D := \{(x_i, y_i)\}_{i=1}^{N_n} \in \mathcal{G}_n$ and a classifier $f$. Let $\beta(x_i) := \mathbb{E}_{Y|X=x_i}[\mathbf{1}(Y(x_i) = f(x_i))]$ be an expected accuracy of $f$ at $x_i$, which is our goal to estimate. Then, due to realization of a single label at each point, the observed accuracy $\hat{\beta}(x_i) := \mathbf{1}(y_i = f(x_i))$ is a random sample from the Bernoulli distribution with parameter $\beta(x_i)$ that has a variance of $\sigma_{x_i}^2 = \beta(x_i)(1 - \beta(x_i))$. Note that this holds when $x_i \neq x_j$ for $i \neq j$, which is the case for most machine learning scenarios. Under this setting, we show the sufficient condition that the maximum likelihood estimator (MLE) of the group accuracy outperforms the MLE of the individual accuracy.

**Proposition 4.1.** *Let $\hat{\beta}^{(id)}$ and $\hat{\beta}^{(gr)}$ be MLEs of individual and group accuracies. Then, $\hat{\beta}^{(gr)}$ has a lower expected mean-squared error than $\hat{\beta}^{(id)}$ if $\frac{1}{4}\left(\max_{x' \in \mathcal{G}_n}\beta(x') - \min_{x' \in \mathcal{G}_n}\beta(x')\right)^2 \leq \frac{N_n-1}{N_n}\bar{\sigma}^2 = \frac{N_n-1}{N_n^2}\sum_{i=1}^{N_n}\beta(x_i)(1 - \beta(x_i))$ where $\bar{\sigma}^2 = \frac{1}{N_n}\sum_{i=1}^{N_n}\sigma_{x_i}^2$.*

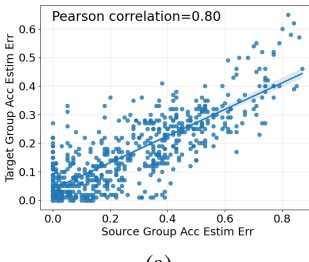 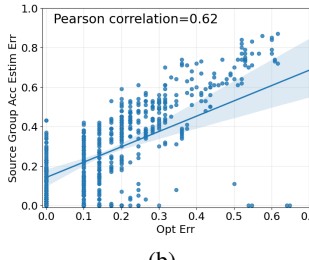 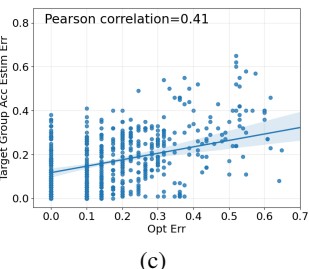

|  |  |  |
|:--:|:--:|:--:|
| (a) | (b) | (c) |

Figure 2: Illustration of correlations between $\epsilon_{opt}(w^\dagger(n))$ and the source and target group accuracy estimation errors. Each point corresponds to an IW and the values are measured on the OfficeHome dataset (720 IWs in total). See Appendix E.7 for more detailed discussions and analyses.

The proof is based on bias-variance decomposition and the Popoviciu's inequality (Popoviciu, 1965), which is given in Appendix A. While $\bar{\sigma}^2$ is fixed, we can reduce the term $\max_{x \in \mathcal{G}_n} \beta(x) - \min_{x \in \mathcal{G}_n} \beta(x)$ through a careful group construction that we discuss in Section 4.2. We also note that the sufficient condition tends to be loose (e.g. $\max_{x \in \mathcal{G}_n} \beta(x) - \min_{x \in \mathcal{G}_n} \beta(x) \le 0.8$ when $\frac{1}{N_n} \sum_{i=1}^{N_n} \beta(x_i) = 0.8$; cf. Figure A1). This means that the group accuracy estimator is statistically more favorable than the individual accuracy estimator in various cases.

## 4.2 IMPORTANCE WEIGHT ESTIMATION AND GROUP CONSTRUCTION

Our goal is to obtain IW $\hat{w}$ that leads to $\alpha_T(\mathcal{G}_n; w^*) \approx \alpha_T(\mathcal{G}_n; \hat{w})$ for $n \in [M]$. The proposed method is based on a CI estimation method developed for producing the PAC prediction set discussed in Section 3 (Park et al., 2022)[1]. Specifically, given the CI of binned IWs $\{\Phi_i\}_{i \in B}$ in (2), our goal is to find binned IWs $\{w_i \in \Phi_i\}_{i \in [B]}$ that give an accurate group accuracy estimation. We let $\tilde{w}(x) := w_{I^{(B)}(x)}$ be the induced IW estimation from binned IWs where $I^{(B)}$ is the partition in (1).

Our idea for accurately estimating the "target" group accuracy with IW estimator $\tilde{w}$ is to define two estimators for the "source" group accuracy defined in (4), with one estimator dependent on a target accuracy estimate, and to encourage the two estimators to agree with each other. This approach can be validated because *the target accuracy estimation error of $\tilde{w}$ can be upper bounded by its source accuracy estimation error*; that is,

$$|\alpha_T(\mathcal{G}_n; w^*) - \alpha_T(\mathcal{G}_n; \tilde{w})| = |\mathbb{E}_{p_T}\left[\tilde{w}(X)\left(\frac{1}{w^*(X)} - \frac{1}{\tilde{w}(X)}\right)\mathbf{1}(Y = \hat{Y})|X \in \mathcal{G}_n\right]|\frac{P(X_S \in \mathcal{G}_n)}{P(X_T \in \mathcal{G}_n)}$$

$$\le \tilde{w}_n^{(ub)} \cdot |\alpha_S(\mathcal{G}_n; w^*) - \alpha_S(\mathcal{G}_n; \tilde{w})|\left(\frac{P(X_T \in \mathcal{G}_n)}{P(X_S \in \mathcal{G}_n)}\right)^2 \quad (5)$$

where $\tilde{w}_n^{(ub)} = \max_{x \in Supp(p_T(\cdot|X \in \mathcal{G}_n))} \tilde{w}(x)$, $\alpha_S(\mathcal{G}_n; \tilde{w})$ is obtained by replacing $w^*$ with $\tilde{w}$ in (4), and the bound is tight when $\tilde{w}(x) = \tilde{w}_n^{(ub)}$ for all $x \in Supp(p_T(\cdot|X \in \mathcal{G}_n))$. Under a loose bound or large values of $\tilde{w}_n^{(ub)}$ and $P(X_T \in \mathcal{G}_n)/P(X_S \in \mathcal{G}_n)$ in (5), reducing $|\alpha_S(\mathcal{G}_n; w^*) - \alpha_S(\mathcal{G}_n; \tilde{w})|$ may not effectively reduce $|\alpha_T(\mathcal{G}_n; w^*) - \alpha_T(\mathcal{G}_n; \tilde{w})|$. In our empirical verification with 720 different IWs (Figure 2(a)), we observe that an IW that has small $|\alpha_S(\mathcal{G}_n; w^*) - \alpha_S(\mathcal{G}_n; \tilde{w})|$ highly likely achieves lower $|\alpha_T(\mathcal{G}_n; w^*) - \alpha_T(\mathcal{G}_n; \tilde{w})|$ compared to others, which resolves potential issues associated with the loose bound or large values of $\tilde{w}_n^{(ub)}$ and $P(X_T \in \mathcal{G}_n)/P(X_S \in \mathcal{G}_n)$.

To develop the first estimator, note that we can reliably approximate the source group accuracy with $\mathcal{D}_S$ by Monte-Carlo estimation with the error of $\mathcal{O}(1/\sqrt{|\mathcal{G}_n(\mathcal{D}_S)|})$ where $\mathcal{G}_n(\mathcal{D}_S) := \{(x_k, y_k) \in \mathcal{D}_S : x_k \in \mathcal{G}_n\}$; we denote the Monte-Carlo estimate as $\hat{\alpha}_S^{(MC)}(\mathcal{G}_n) = \hat{\mathbb{E}}_{p_S}[\mathbf{1}(Y = \hat{Y})|X \in \mathcal{G}_n]$.

---

[1]However, we want to emphasize that IW-GAE can be applied to any valid CI estimators (cf. Appendix B.2). In addition, we show that IW-GAE outperforms state-of-the-art methods even a naive CI estimator that sets minimum and maximum values of binned IWs as CIs in Appendix E.5.

Based on (4), we define a second estimator for $\alpha_S(\mathcal{G}_i; w^*)$, as a function of binned IWs $\{w_i\}_{i \in [B]}$, by assuming $\mathbb{E}_{p_{T_{Y|x}}}[\mathbf{1}(Y(x) = \hat{Y}(x))] = \hat{\alpha}_T(\mathcal{G}_n; \{w_i\}_{i \in [B]})$ for all $x \in \mathcal{G}_n$:

$$\hat{\alpha}_S^{(IW)}(\mathcal{G}_n; \{w_i\}_{i \in [B]}) := \frac{\hat{P}(X_T \in \mathcal{G}_n)}{\hat{P}(X_S \in \mathcal{G}_n)} \cdot \hat{\mathbb{E}}_{p_T}\left[\frac{\hat{\alpha}_T(\mathcal{G}_n; \{w_i\}_{i \in [B]})}{\tilde{w}(X)} \Big| X \in \mathcal{G}_n\right]$$

$$= \hat{\mathbb{E}}_{p_T}\left[\frac{1}{\tilde{w}(X)} \Big| X \in \mathcal{G}_n\right] \hat{\mathbb{E}}_{p_S}[\mathbf{1}(Y = \hat{Y})\tilde{w}(X) | X \in \mathcal{G}_n] \quad (6)$$

where $\hat{\alpha}_T(\mathcal{G}_n; \{w_i\}_{i \in [B]})$ is an empirical estimate of the target accuracy with $\{w_i\}_{i \in [B]}$ in (3), $\hat{P}(X_T \in \mathcal{G}_n) := \hat{E}_{p_T}[\mathbf{1}(X \in \mathcal{G}_n)]$, and $\hat{P}(X_S \in \mathcal{G}_n) := \hat{E}_{p_S}[\mathbf{1}(X \in \mathcal{G}_n)]$.

We aim to formulate an optimization problem to choose binned IWs from CIs such that $\min_{\{w_i \in \Phi_i\}_{i \in [B]}}(\hat{\alpha}_S^{(IW)}(\mathcal{G}_n; \{w_i\}_{i \in [B]}) - \hat{\alpha}_S^{(MC)}(\mathcal{G}_n))^2$. However, note that $\hat{\alpha}_S^{(IW)}(\mathcal{G}_n; \{w_i\}_{i \in [B]})$ in (6) is non-convex with respect to $w_i$'s (see Appendix A.2 for the derivation), which is in general not effectively solvable with optimization methods (Jain et al., 2017). Therefore, we introduce a relaxed reformulation of (6) by separating binned IWs for source and target, which introduces coordinatewise convexity. Specifically, we redefine the estimator in (6) as

$$\hat{\alpha}_S^{(IW)}(\mathcal{G}_n; \{w_i^{(S)}, w_i^{(T)}\}_{i \in [B]}) := \hat{\mathbb{E}}_{p_T}\left[\frac{1}{\tilde{w}^{(T)}(X)} \Big| X \in \mathcal{G}_n\right] \hat{\mathbb{E}}_{p_S}\left[\mathbf{1}(Y = \hat{Y})\tilde{w}^{(S)}(X) | X \in \mathcal{G}_n\right]$$
$$(7)$$

where $\tilde{w}^{(S)}(X) := w_{I^{(B)}(X)}^{(S)}$ and $\tilde{w}^{(T)}(X) := w_{I^{(B)}(X)}^{(T)}$. Then, we encourage agreements of $w_i^{(S)}$ and $w_i^{(T)}$ for $i \in [B]$ through constraints. Specifically, for each group $\mathcal{G}_n$, we find binned IWs $w^\dagger(n) \in \mathbb{R}_+^{2B}$ by solving the following optimization:

$$w^\dagger(n) \in \underset{\{w_i^{(S)}, w_i^{(T)}\}_{i \in [B]}}{\arg\min} \left(\hat{\alpha}_S^{(MC)}(\mathcal{G}_n) - \hat{\alpha}_S^{(IW)}(\mathcal{G}_n; \{w_i^{(S)}, w_i^{(T)}\}_{i \in [B]})\right)^2 \quad (8)$$

$$\text{s.t.} \quad w_i^{(S)} \in \Phi_i, \quad \text{for } i \in [B] \quad (9)$$

$$w_i^{(T)} \in \Phi_i, \quad \text{for } i \in [B] \quad (10)$$

$$\| w_i^{(T)} - w_i^{(S)} \|_2^2 \leq \delta^{(tol)} \quad \text{for } i \in [B] \quad (11)$$

$$\left|\hat{\mathbb{E}}_{p_S}[\tilde{w}^{(S)}(X) | X \in \mathcal{G}_n] - \frac{\hat{P}(X_T \in \mathcal{G}_n)}{\hat{P}(X_S \in \mathcal{G}_n)}\right| \leq \delta^{(prob)} \quad (12)$$

$$\left|\hat{\mathbb{E}}_{p_T}[1/\tilde{w}^{(T)}(X) | X \in \mathcal{G}_n] - \frac{\hat{P}(X_S \in \mathcal{G}_n)}{\hat{P}(X_T \in \mathcal{G}_n)}\right| \leq \delta^{(prob)} \quad (13)$$

where $\delta^{(tol)}$ and $\delta^{(prob)}$ are small constants. Box constraints (9) and (10) ensure that the obtained solution is in the CI, which bounds the estimation error of $w_i^{(S)}$ and $w_i^{(T)}$ by $|\Phi_i|$. This can also bound the target group accuracy estimation error as $|\alpha_T(\mathcal{G}_n; w^*) - \alpha_T(\mathcal{G}_n; \{w_i^{(S)}\}_{i \in B})| \leq \max_{b \in [B]} |\Phi_b| P(X_S \in \mathcal{G}_n)/P(X_T \in \mathcal{G}_n)$. Constraint (11) corresponds to the relaxation for removing non-convexity of the original objective, and setting $\delta^{(tol)} = 0$ recovers the original objective. Constraints (12) and (13) are based on the equalities that the true IW $w^*(\cdot)$ satisfies: $\mathbb{E}_{p_S}[w^*(X) | X \in \mathcal{G}_n] = \frac{P(X_T \in \mathcal{G}_n)}{P(X_S \in \mathcal{G}_n)}$ and $\mathbb{E}_{p_T}[1/w^*(X) | X \in \mathcal{G}_n] = \frac{P(X_S \in \mathcal{G}_n)}{P(X_T \in \mathcal{G}_n)}$.

Since the above optimization problem is a constrained nonlinear optimization problem with box constraints, we solve it through sequential least square programming (Kraft, 1988). Note that the objective (8) is convex with respect to a block $(w_1^{(S)}, w_2^{(S)}, \cdots, w_B^{(S)})$ and a block $(w_1^{(T)}, w_2^{(T)}, \cdots, w_B^{(T)})$, but not jointly convex. Therefore, using a quasi-Newton method can guarantee only convergence to a local optimum (Nocedal and Wright, 1999). Nevertheless, due to constraints (12) and (13), the asymptotic convergence $(w^\dagger(n))_i \to w_i^*$ and $(w^\dagger(n))_{i+B} \to w_i^*$ as $N^{(S)} \to \infty$ and $N^{(T)} \to \infty$ can be trivially guaranteed because $|\Phi_i| \to 0$ for $i \in [B]$ (Thulin, 2014).

The above optimization problem can be thought of as aiming to estimate the truncated IW $w(x|X \in \mathcal{G}_n) := \frac{p_T(x|X \in \mathcal{G}_n)}{p_S(x|X \in \mathcal{G}_n)}$ for each $\mathcal{G}_n$ that can induce an accurate source group accuracy estimator. However, the objective in (8) does not measure the source group accuracy estimation error. In the following proposition, we show that the above optimization *minimizes the upper bound of the source group accuracy estimation error*, thereby the target group accuracy estimation error due to (5).

**Proposition 4.2** (Upper bound of source group accuracy estimation error). *Let $w^\dagger(n)$ be a solution to the optimization problem for $\mathcal{G}_n$ defined in (8)-(13) with $\delta^{(tol)} = 0$ and $\delta^{(prob)} = 0$. Let $\epsilon_{opt}(w^\dagger(n)) := \left( \hat{\alpha}_S^{(MC)}(\mathcal{G}_n) - \hat{\alpha}_S^{(IW)}(\mathcal{G}_n; w^\dagger(n)) \right)^2$ be the objective value. For $\tilde{\delta} > 0$, the following inequality holds with probability at least $1 - \tilde{\delta}$:*

$$|\alpha_S(\mathcal{G}_n; w^*) - \alpha_S(\mathcal{G}_n; w^\dagger(n))| \leq \epsilon_{opt}(w^\dagger(n)) + \epsilon_{stat} + |\alpha_S^{(IW)}(\mathcal{G}_n; w^\dagger(n)) - \alpha_S(\mathcal{G}_n; w^\dagger(n))| \tag{14}$$

$$\leq \epsilon_{opt}(w^\dagger(n)) + \epsilon_{stat} + IdentBias(w^\dagger(n); \mathcal{G}_n) \tag{15}$$

*where $IdentBias(w^\dagger(n); \mathcal{G}_n) = \frac{P(X_T \in \mathcal{G}_n)}{2P(X_S \in \mathcal{G}_n)} (\mathbb{E}_{p_T}[(\mathbf{I}(Y(X) = \hat{Y}(X)) - \alpha_T(\mathcal{G}_n; w^*))^2 | X \in \mathcal{G}_n] + \frac{1}{\underline{w}^\dagger(n)^2})$ and $\epsilon_{stat} \in \mathcal{O}(\log(1/\tilde{\delta}))/\sqrt{|\mathcal{G}_n(\mathcal{D}_S)|}$ for $\underline{w}^\dagger(n) := \min_{i \in [2B]}\{w_i^\dagger(n)\}$.*

The proof is based on the Cauchy-Schwarz inequality, which is provided in Appendix A. The bound in (14) is tight when values of $\alpha_S(\mathcal{G}_n; w^*)$, $\hat{\alpha}_S^{(MC)}(\mathcal{G}_n)$, $\hat{\alpha}_S^{(IW)}(\mathcal{G}_n; w^\dagger(n))$, $\alpha_S^{(IW)}(\mathcal{G}_n; w^\dagger(n))$, and $\alpha_S(\mathcal{G}_n; w^\dagger(n))$ are monotonic. Proposition 4.2 shows that we can reduce the source accuracy estimation error by reducing $\epsilon_{opt}(w^\dagger(n))$ by solving the optimization problem. However, a large value of $\epsilon_{stat} + |\alpha_S^{(IW)}(\mathcal{G}_n; w^\dagger(n)) - \alpha_S(\mathcal{G}_n; w^\dagger(n))|$ or a looseness of (14) could significantly decrease the effectiveness of IW-GAE. In this regard, we analyze the relationships between $\epsilon_{opt}(w^\dagger(n))$ and two group accuracy estimation errors in Figures 2(b) and 2(c). In the empirical analyses, it turns out that reducing $\epsilon_{opt}(w^\dagger(n))$ can effectively reduce the group accuracy estimation in both source and target domains, which advocates the direction of IW-GAE.

For designining the group assignment function $I^{(g)}$, we note that $IdentBias(w^\dagger(n); \mathcal{G}_n)$ can be reduced by decreasing the variance of the correctness within the group (cf. Proposition A.1). Thus, we group examples by the maximum value of the softmax output as in Guo et al. (2017) based on strong empirical evidence that the maximum value of the softmax output is highly correlated with accuracy in UDA (Wang et al., 2020). In addition, we introduce a learnable temperature scale parameter for target samples for adjusting the sharpness of the softmax output for target predictions inspired by results that an overall scale of the maximum value of the softmax output significantly varies from one domain to another (Yu et al., 2022). Specifically, groups in the source and target domains are defined as $\mathcal{G}_n(\mathcal{D}_S) := \{x_i \in \mathcal{D} | \frac{n-1}{M} \leq (\phi(g(x_i)))_j \leq \frac{n}{M}, j \in m(x_i)\}$ and $\mathcal{G}_n^{(t)}(\mathcal{D}_T) := \{x_i \in \mathcal{D}_T | \frac{n-1}{M} \leq (\phi(g(x_i)/t))_j \leq \frac{n}{M}, j \in m(x_i)\}$ for $n \in [M]$ where $g : \mathcal{X} \to \mathbb{R}^K$ is the logit function of a neural network, $\phi$ is the $K$-dimensional softmax function, and $m(x_i) := \arg\max_{k \in [K]} (\phi(g(x_i)))_k$.

We note that introducing the temperature scaling parameter results in the nested optimization of $\min_{t \in \mathcal{T}} \left\{ \text{optimization problem in (8)-(13) with } \mathcal{G}_n(\mathcal{D}_S) \text{ and } \mathcal{G}_n^{(t)}(\mathcal{D}_T) \right\}$ (see Appendix C.1 for the complete optimization form and Algorithm 1 for pseudocode). Based on the facts that a group separation is not sensitive to small changes in the temperature and the inner optimization is not smooth with respect to $t$, we use a discrete set for $\mathcal{T} := \{t_1, t_2, \cdots, t_n\}$. We note that the inner optimization problem is readily solvable, so the discrete optimization can be performed without much additional computational overhead.

Finally, for tightening the upper bounds in (5) and (15), we bound the maximum and minimum values of an IW estimation by $w^{(ub)}$ and $w^{(lb)}$, which is a common technique in IW-based estimations (Wang et al., 2020; Park et al., 2022). We note that bounding an IW estimation value only affects the estimator, which does not affect any theoretical guarantee based on properties of true IWs such as Proposition 4.2 and inequalities in (5) and (15). However, we remark that bounding an IW estimation may increase $\epsilon_{opt}(w^\dagger(n))$ due to reduced search space, i.e., $w_i^{(S)} \in \Phi_i \cap [w^{(lb)}, w^{(ub)}]$ in (9) and $w_i^{(T)} \in \Phi_i \cap [w^{(lb)}, w^{(ub)}]$ in (10), although we observe that it works effectively in practice.

## 5 EXPERIMENTS

We evaluate IW-GAE on model calibration and selection tasks. Since both tasks are based on UDA classification tasks, we first provide the common setup and task-specific setup such as the baselines and evaluation metrics in the corresponding sections. For all experiments in this section, we evaluate

our method on the popular Office-Home (Venkateswara et al., 2017) dataset, which contains around 15,000 images of 65 categories from four domains (Art, Clipart, Product, Real-World).

**A base model** is required for implementing the baseline methods and IW-GAE, which serve as the test objectives for the model calibration and selection tasks. We consider maximum mean discrepancy (MDD; (Zhang et al., 2019)) with ResNet-50 (He et al., 2016) as the backbone neural network, which is the most popular high-performing UDA method. MDD aims to learn domain invariant representation while learning a classification task in the source domain. In implementation, we use the popular open source project Transfer Learning Library (Jiang et al., 2020). We use the default hyperparameters in all experiments. Further details are explained in Appendix D.

**IW estimation** is required for implementing baseline methods and construct bins for estimating the CI of the IW. We adopt a linear logistic regression model on top of the neural network's representation as the discriminative learning-based estimation, following Wang et al. (2020). Specifically, it first upsamples from one domain to make $|\mathcal{D}_S| = |\mathcal{D}_T|$, and then it labels samples with the domain index: $\{(h(x), 1)|x \in \mathcal{D}_T\}$ and $\{(h(x), 0)|x \in \mathcal{D}_S\}$ where $h$ is the feature map of the neural network. Then, logistic regression is trained with a quasi-Newton method until convergence.

## 5.1 MODEL CALIBRATION PERFORMANCE

**Setup & Metric** In this experiment, our goal is to match the confidence of a prediction to its expected accuracy in the target domain. Following the standard (Guo et al., 2017; Park et al., 2020; Wang et al., 2020), we use expected calibration error (ECE) on the test dataset as a measure of calibration performance. The ECE measures the average absolute difference between the confidence and accuracy of binned groups, which is defined as $ECE(\mathcal{D}_T) = \sum_{n=1}^{m} \frac{|\mathcal{G}_n|}{|\mathcal{D}_T|}|\hat{\text{Acc}}(\mathcal{G}_n(\mathcal{D}_T)) - \hat{\text{Conf}}(\mathcal{G}_n(\mathcal{D}_T))|$ where $\hat{\text{Acc}}(\mathcal{G}_n(\mathcal{D}_T))$ is the average accuracy in $\mathcal{G}_n(\mathcal{D}_T)$ and $\hat{\text{Conf}}(\mathcal{G}_n(\mathcal{D}_T))$ is the average confidence in $\mathcal{G}_n(\mathcal{D}_T)$. We use $M = 15$ following the standard value (Guo et al., 2017; Wang et al., 2020).

**Baselines** We consider the following five different baselines: The *vanilla* method uses a maximum value of the softmax output as the confidence of the prediction. We also consider temperature scaling-based methods that adjust the temperature parameter by maximizing the following calibration measures: *Temperature scaling (TS)* (Guo et al., 2017): the log-likelihood on the source validation dataset; *IW temperature scaling (IW-TS)*: the log-likelihood on the importance weighted source validation dataset; *Calibrated prediction with covariate shift (CPCS)*: the Brier score (Brier, 1950) on the importance weighted source validation dataset; *TransCal* (Wang et al., 2020): the ECE on the importance weighted source validation dataset with a bias and variance reduction technique. These methods also use a maximum value of the (temperature-scaled) softmax output as the confidence.

**Results** As shown in Table 1, IW-GAE achieves the best average ECEs across different base models. For individual domains, IW-GAE achieves the best ECE among 11 out of the 12 cases. We note that IW-Mid, which selects the middle point in the CI as IW estimation and originates herein as a simple replacement of a classic IW estimation technique (Bickel et al., 2007) by a recently proposed CI estimator (Park et al., 2022), is a strong baseline, outperforming other baselines. IW-GAE improves this strong baseline for every case. This shows that the process of reducing $\epsilon_{opt}(w^\dagger(n))$ reduces the group accuracy estimation error in the target domain, which is consistent with the result in Proposition 4.2. Finally, we show that the effectiveness of IW-GAE can be generalized to large-scale datasets (VisDa-2017 and DomainNet) in Appendix E.1 and different base models (conditional adversarial domain adaptation (Long et al., 2018) and maximum classifier discrepancy (Saito et al., 2018)) in E.2, outperforming state-of-the-art performances by 21%, 31%, 2%, and 5% respectively.

## 5.2 MODEL SELECTION

**Setup & Metric** In this experiment, we perform model selection for choosing the best hyperparameter. To this end, we repeat training the MDD method by changing its key hyperparameter of margin coefficient from 1 to 8 (the default value is 4). After training several models under different values of the margin coefficient, we choose one model based on a model selection criterion. For evaluation, we compare the test target accuracy of the chosen model under different model selection methods.

**Baselines** We consider three baselines that evaluate the model's performance in terms of the following criterion: *Vanilla*: the minimum classification error on the source validation dataset; *Importance weighted cross validation (IWCV)* (Sugiyama et al., 2007): the minimum importance-weighted

| Task | Method | Ar-Cl | Ar-Pr | Ar-Rw | Cl-Ar | Cl-Pr | Cl-Rw | Pr-Ar | Pr-Cl | Pr-Rw | Rw-Ar | Rw-Cl | Rw-Pr | Avg |
|---|---|---|---|---|---|---|---|---|---|---|---|---|---|---|
| Model calibration | Vanilla | 40.61 | 25.62 | 15.56 | 33.83 | 25.34 | 24.75 | 33.45 | 38.62 | 16.76 | 23.37 | 36.51 | 14.01 | 27.37 |
| | TS | 35.86 | 22.84 | 10.60 | 28.24 | 20.74 | 20.06 | 32.47 | 37.20 | 14.89 | 18.36 | 34.62 | 12.28 | 24.01 |
| | CPCS | 22.93 | 22.07 | 10.19 | 26.88 | 18.36 | 14.05 | 28.28 | 29.20 | 12.06 | 15.76 | 26.54 | 11.14 | 19.79 |
| | IW-TS | 32.63 | 22.90 | 11.27 | 28.05 | 19.65 | 18.67 | 30.77 | 38.46 | 15.10 | 17.69 | 32.20 | 11.77 | 23.26 |
| | TransCal | 33.57 | 20.27 | **8.88** | 26.36 | 18.81 | 18.42 | 27.35 | 29.86 | 10.48 | 16.17 | 29.90 | 10.00 | 20.84 |
| | IW-Mid | 23.25 | 31.62 | 12.99 | 17.15 | 18.71 | 9.23 | 27.75 | 30.35 | 9.02 | 13.64 | 26.32 | 10.60 | 19.22 |
| | IW-GAE | **12.78** | **4.70** | 12.93 | **7.52** | **4.42** | **4.11** | **9.50** | **17.49** | **8.40** | **7.62** | **9.52** | **8.14** | **8.93** |
| | Oracle | 10.45 | 10.72 | 6.47 | 8.10 | 7.62 | 6.55 | 11.88 | 9.39 | 5.93 | 7.54 | 10.72 | 5.70 | 8.42 |
| Model selection | Vanilla | 53.31 | **70.96** | 77.44 | 59.70 | 65.17 | 69.96 | 50.95 | 57.07 | 74.75 | 68.81 | 57.11 | 80.13 | 65.45 |
| | IWCV | 53.24 | 69.61 | 72.50 | 59.70 | 65.17 | 67.50 | 57.07 | 55.21 | 74.75 | 68.81 | 58.51 | 80.13 | 65.18 |
| | DEV | 53.31 | 70.72 | 77.44 | 59.79 | 67.99 | 69.96 | 57.07 | 52.50 | 77.12 | **70.50** | 53.38 | 82.27 | 66.00 |
| | IW-Mid | 54.13 | 69.27 | **78.47** | **61.48** | 68.03 | **71.06** | 59.99 | 55.21 | 78.79 | **70.50** | 57.11 | 83.10 | 67.26 |
| | IW-GAE | **54.34** | **70.96** | **78.47** | **61.48** | **69.93** | **71.06** | **62.79** | **55.21** | **78.79** | **70.50** | **58.51** | **83.31** | **67.95** |
| | Lower bound | 52.51 | 69.27 | 72.50 | 59.70 | 65.17 | 67.50 | 57.07 | 50.95 | 74.75 | 68.81 | 50.90 | 80.13 | 64.10 |
| | Oracle | 54.34 | 70.96 | 78.47 | 61.48 | 69.93 | 71.06 | 62.79 | 55.21 | 78.79 | 71.32 | 58.51 | 83.31 | 68.01 |

Table 1: Model calibration and selection benchmark results of MDD with ResNet-50 on Office-Home. We repeat experiments for ten times and report the average value. For the model calibration, the numbers indicate the mean ECE with boldface for the minimum mean ECE. For the model selection, the numbers indicate the mean test accuracy of selected model with boldface for the maximum mean test accuracy. For the model calibration task, Oracle is obtained by applying TS with labeled test samples in the target domain. For the model selection task, lower bound and Oracle indicate the accuracy of the models with the worst and best test accuracy, respectively.

classification error on the source validation dataset; *Deep embedded validation (DEV))* (You et al., 2019): the minimum deep embedded validation risk on the source validation dataset.

**Results** Table 1 shows that model selection with IW-GAE achieves the best average accuracy, improving state-of-the-art by 18% in terms of the relative scale of lower and upper bounds of accuracy. Specifically, IW-GAE achieves the best performance in all cases. We also note that IW-Mid performs the model selection task surprisingly well. This means that, on average, the true IW could be located near the middle point of the CI, while the exact location varies from one group to another. Note that plain IWCV does not improve the vanilla method on average, which could be due to the inaccurate estimation of the IW by the discriminative learning-based approach. In this sense, IW-GAE has an advantage of depending less on the performance of the IW estimator since the estimated value is used to construct bins for the CI, and then the exact value is found by solving the separate optimization problem. We also remark that our experimental results reveal *dangers of the current practice of using the vanilla method or IWCV in model selection in UDA*. Finally, in Appendix E.3, we show that IW-GAE effectively solves another model selection task of choosing the best checkpoint, outperforming state-of-the-art performance by 9%.

### 5.3 QUALITATIVE EVALUATION, ABLATION STUDY, AND SENSITIVITY ANALYSIS

In Appendix E.4, we qualitatively evaluate IW-GAE by visually comparing the group accuracy estimation and the average group accuracy, which shows an accurate estimation ability of IW-GAE. In Appendix E.5, we show that the group construction criterion and nested optimization with temperature scaling developed in Section 4.2 work effectively for IW-GAE. In Appendix E.6, a sensitivity analysis shows that IW-GAE consistently performs well even under large changes in the number of bins $B$ and the number of accuracy groups $M$.

## 6 CONCLUSION

In this work, we formulate an optimization problem to choose IW estimation from its CI for accurately estimating group accuracy. Specifically, we define a Monte-Carlo estimator and an IW-based estimator of group accuracy in the source domain and find the IW that makes the two estimators close to each other. Solving the optimization problem not only *reduces the source group accuracy estimation error* but also *leads to an accurate group accuracy estimation in the target domain*. We show that our method achieves state-of-the-art performances in both model calibration and selection tasks in UDA across a wide range of benchmark problems. We believe that the impressive performance gains by our method show a promising future direction of research, which is orthogonal to improving the transferability of accuracy–the main focus in the UDA literature. Finally, we note that all IW-based methods (CPCS, IW-TS, TransCal, IW-GAE) fail to improve the standard method in the i.i.d. scenario in our experiments with pre-trained large-language models (XLM-R (Conneau et al., 2019) and GPT-2 (Solaiman et al., 2019)). We conjecture that these models are less subject to the distribution shifts due to massive amounts of training data that may include the target domain datasets, so applying the methods in the i.i.d. setting can work effectively. In this regard, we leave the following important research questions: "Are IW-based methods less effective, or even detrimental, under mild distribution shifts?" and "Can we develop methods that work well under all levels of distribution shifts?"

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

# A  PROOF OF CLAIMS

## A.1  PROOF OF PROPOSITION 4.1

*Proof.* The proof consists of three parts: 1) decomposition of the expected mean-square error of an estimator $g(x)$; 2) deriving MLEs of individual and group accuracies; 3) constructing a sufficient condition.

**1) Bias-variance decomposition of the expected mean-square error** The expected mean-square error of an estimator $g(x)$ for $\beta(x)$ at $x_i \in \mathcal{G}_n$ with respect to the realization of a label $y_i \sim Y|x$ can be decomposed by

$$\mathbb{E}_D[(\hat{\beta}(x_i) - g(x_i))^2] = Var_D(g(x_i; D)) + (Bias_D(g(x_i; D)))^2 + \sigma_{x_i}^2 \qquad (16)$$

where $Var_D(g(x_i; D)) := \mathbb{E}_D[(g(x_i; D) - \mathbb{E}_D[g(x_i; D)])^2]$ is the variance of the estimator and $Bias_D(g(x_i; D)) := \mathbb{E}_D[g(x_i; D)] - \beta(x)$ is the bias of the estimator.

**2) MLEs of individual and group accuracy estimators** For an individual accuracy estimator $\hat{\beta}^{(id)}(x; D)$ that predicts an accuracy for each sample $x$ given $D$, an MLE estimator is $\hat{\beta}^{(id)}(x) = \hat{\beta}(x)$. This estimator is unbiased because $\mathbb{E}_D(\hat{\beta}^{(id)}(x; D)) = \beta(x)$ for each $x \in \mathcal{G}_n$. Therefore, this estimator has the average of expected errors

$$\frac{1}{N_n} \sum_{k=1}^{N_n} \mathbb{E}_D[(\hat{\beta}(x_k) - \hat{\beta}^{(id)}(x_k; D))^2] = \bar{\sigma}^2 + \bar{\sigma}^2 \qquad (17)$$

where $\bar{\sigma}^2 := \frac{1}{N_n} \sum_{i=1}^{N_n} \sigma_{x_i}^2$.

For a group accuracy estimator $\hat{\beta}^{(gr)}(x; D)$ that predicts the same group accuracy estimate for all $x \in \mathcal{G}_n$, an MLE estimator can be defined by $\hat{\beta}^{(gr)}(x; D) = \frac{1}{N_n} \sum_{i=1}^{N_n} \hat{\beta}(x_i)$, which is a biased estimator because $\mathbb{E}_D(\hat{\beta}^{(gr)}(x; D)) = \frac{1}{N_n} \sum_{i=1}^{N_n} \beta(x_i)$ for each $x \in \mathcal{G}_n$. Therefore, this estimator has the average of expected errors

$$\frac{1}{N_n} \sum_{k=1}^{N_n} \mathbb{E}_D[(\hat{\beta}(x_k) - \hat{\beta}^{(gr)}(x_k))^2] = \frac{1}{N_n} \bar{\sigma}^2 + \frac{1}{N_n} \sum_{k=1}^{N_n} \left( \frac{1}{N_n} \sum_{i=1}^{N_n} \beta(x_i) - \beta(x_k) \right)^2 + \bar{\sigma}^2 \qquad (18)$$

$$= \frac{1}{N_n} \bar{\sigma}^2 + Var(\beta; D) + \bar{\sigma}^2 \qquad (19)$$

where $Var(\beta; D)$ is the variance of the accuracy in group $\mathcal{G}_n$.

**3) Sufficient condition** Given (17) and (19), the Popoviciu's inequality (Popoviciu, 1965) provides a sufficient condition for the group accuracy estimator $\hat{\beta}^{(gr)}$ to have a lower expected mean-squared error than the individual accuracy estimator $\hat{\beta}^{(id)}$ as follows:

$$Var(\beta; D) \leq \frac{1}{4} \left( \max_{x' \in \mathcal{G}_n} \beta(x') - \min_{x' \in \mathcal{G}_n} \beta(x') \right)^2 \leq \frac{N_n - 1}{N_n} \bar{\sigma}^2 = \frac{N_n - 1}{N_n} \left( \frac{1}{N_n} \sum_{i=1}^{N_n} \beta(x_i)(1 - \beta(x_i)) \right) \qquad (20)$$

where the equality comes from $\bar{\sigma}^2 = \frac{1}{N_n} \sum_{i=1}^{N_n} \sigma_{x_i}^2$ with $\sigma_{x_i}^2$ is the variance of the Bernoulli distribution with a parameter $\beta(x_i)$. $\qquad \square$

## A.2  NON-CONVEXITY OF THE OPTIMIZATION PROBLEM

Let $\mathcal{G}_n(\mathcal{D}_S) := \{(x_k, y_k) \in \mathcal{D}_S : x_k \in \mathcal{G}_n\}$ and $\mathcal{G}_n(\mathcal{D}_T) := \{x_k \in \mathcal{D}_T : x_k \in \mathcal{G}_n\}$ for $n \in [M]$. By elementary algebra, we obtain the following

$$\hat{\alpha}_S^{(IW)}(\mathcal{G}_n; \{w_i\}_{i \in [B]}) = \left( \frac{1}{|\mathcal{G}_n(\mathcal{D}_T)|} \sum_{(x,y)) \in \mathcal{G}_n(\mathcal{D}_T)} \frac{1}{w(x)} \right) \left( \frac{1}{|\mathcal{G}_n(\mathcal{D}_S)|} \sum_{x \in \mathcal{G}_n(\mathcal{D}_S)} \mathbf{1}(y = \hat{Y}(x))w(x) \right) \qquad (21)$$

$$= \left( \frac{1}{|\mathcal{G}_n(\mathcal{D}_T)|} \sum_{i=1}^{B} \frac{a_i}{w_i} \right) \left( \frac{1}{|\mathcal{G}_n(\mathcal{D}_S)|} \sum_{i=1}^{B} b_i w_i \right) \qquad (22)$$

where $a_i = |\mathcal{G}_n(\mathcal{D}_T) \cap \mathcal{B}_i|$ and $b_i = |\{(x_k, y_k) \in \mathcal{G}_n(\mathcal{D}_S) \cap \mathcal{B}_i | y_k = \hat{Y}(x_k)\}|$. Therefore, $\hat{\alpha}_S^{(IW)}(\mathcal{G}_n; \{w_i\}_{i \in [B]})$ is non-convex with respect to $w_i$ for $i \in [B]$.

## A.3 PROOF OF PROPOSITION 4.2

*Proof.* By applying triangle inequalities, we get the following inequality:

$$|\alpha_S(\mathcal{G}_n; w^*) - \alpha_S(\mathcal{G}_n; w^\dagger(n))| \leq |\alpha_S(\mathcal{G}_n; w^*) - \hat{\alpha}_S^{(MC)}(\mathcal{G}_n)| + |\hat{\alpha}_S^{(MC)}(\mathcal{G}_n) - \hat{\alpha}_S^{(IW)}(\mathcal{G}_n; w^\dagger(n))|$$
$$+ |\hat{\alpha}_S^{(IW)}(\mathcal{G}_n; w^\dagger(n)) - \alpha_S^{(IW)}(\mathcal{G}_n; w^\dagger(n))| + |\alpha_S^{(IW)}(\mathcal{G}_n; w^\dagger(n)) - \alpha_S(\mathcal{G}_n; w^\dagger(n))|. \quad (23)$$

Note that the first and third terms in the right hand side are coming from the Monte-Carlo approximation, so they can be bounded by $\mathcal{O}(\log(1/\tilde{\delta})/|\mathcal{D}_n^S|)$ with probability at least $1 - \tilde{\delta}$ based on a concentration inequality such as the Hoeffding's inequality. Also, the second term is bounded by the optimization error $\epsilon_{opt}(w^\dagger(n))$. Therefore, it is enough to analyze the fourth term.

The fourth term is coming from the bias of $\mathbb{E}_{T_{Y|X}}[\mathbf{1}(Y(X) = \hat{Y}(X))] = \hat{\alpha}_T(\mathcal{G}_n; w^\dagger(n))$, which we refer to as *the bias of the identical accuracy assumption*. It can be bounded by

$$|\alpha_S^{(IW)}(\mathcal{G}_n; w^\dagger(n)) - \alpha_S(\mathcal{G}_n; w^\dagger(n))| \quad (24)$$

$$= \frac{P(X_T \in \mathcal{G}_n)}{P(X_S \in \mathcal{G}_n)}\left|\mathbb{E}_{p_T}\left[\frac{\mathbf{1}(Y(X) = \hat{Y}(X)) - \hat{\alpha}_T(\mathcal{G}_n; w^\dagger(n))}{w^\dagger(n)(X)}\middle| X \in \mathcal{G}_n\right]\right| \quad (25)$$

$$\leq \frac{P(X_T \in \mathcal{G}_n)}{P(X_S \in \mathcal{G}_n)}\left(\mathbb{E}_{p_T}\left[(\mathbf{1}(Y(X) = \hat{Y}(X)) - \hat{\alpha}_T(\mathcal{G}_n; w^\dagger(n)))^2\middle| X \in \mathcal{G}_n\right]\mathbb{E}_{p_T}\left[\frac{1}{w^\dagger(n)(X)^2}\middle| X \in \mathcal{G}_n\right]\right)^{1/2} \quad (26)$$

$$\leq \frac{P(X_T \in \mathcal{G}_n)}{2P(X_S \in \mathcal{G}_n)}\left(\mathbb{E}_{p_T}\left[(\mathbf{1}(Y(X) = \hat{Y}(X)) - \hat{\alpha}_T(\mathcal{G}_n; w^\dagger(n)))^2\middle| X \in \mathcal{G}_n\right] + \mathbb{E}_{p_T}\left[\frac{1}{w^\dagger(n)(X)^2}\middle| X \in \mathcal{G}_n\right]\right) \quad (27)$$

$$\leq \frac{P(X_T \in \mathcal{G}_n)}{2P(X_S \in \mathcal{G}_n)}\left(\mathbb{E}_{p_T}\left[(\mathbf{1}(Y = \hat{Y}) - \alpha_T(\mathcal{G}_n; w^*))^2\middle| \mathcal{G}_n\right] + \frac{1}{\underline{w}^\dagger(n)^2}\right) \quad (28)$$

where (26) holds due to the Cauchy-Schwarz inequality, (27) holds due to the AM-GM inequality, and $\underline{w}^\dagger(n) := \min_{i \in [2B]}\{w_i^\dagger(n)\}$. $\qquad\square$

## A.4 FORMAL STATEMENT AND PROOF OF PROPOSITION A.1

**Proposition A.1** (Bias-variance decomposition). *Let $\hat{\alpha}_T(\mathcal{G}_n)$ be an estimate for $\alpha_T(\mathcal{G}_n; w^*)$. Then, the bias of the identical accuracy assumption is given by*

$$IdentBias(w^\dagger(n); \mathcal{G}_n) = \frac{P(X_T \in \mathcal{G}_n)}{2P(X_S \in \mathcal{G}_n)}\left(\frac{1}{\underline{w}^\dagger(n)^2} + Bias(\hat{\alpha}_T(\mathcal{G}_n))^2 + Var(\mathbf{1}(Y = \hat{Y})|\mathcal{G}_n)\right) \quad (29)$$

*where $Bias(\hat{\alpha}_T(\mathcal{G}_n)) := |\alpha_T(\mathcal{G}_n; w^*) - \hat{\alpha}_T(\mathcal{G}_n)|$ is the bias of the estimate $\hat{\alpha}_T(\mathcal{G}_n)$ and $Var(\mathbf{1}(Y = \hat{Y})|\mathcal{G}_n) := \mathbb{E}_{p_T}\left(\mathbf{1}(Y = \hat{Y}) - \alpha_T(\mathcal{G}_n; w^*)\right)^2$ is the variance of the correctness of predictions in $\mathcal{G}_n$.*

*Proof.* Based on the proof of Proposition 4.2, it is enough to decompose $\mathbb{E}_{p_T}[(|\mathbf{1}(Y(X) = \hat{Y}(X)) - \hat{\alpha}_T(\mathcal{G}_n)]^2$ as follows

$$\mathbb{E}_{p_T}\left(\mathbf{1}(Y(X) = \hat{Y}(X)) - \hat{\alpha}_T(\mathcal{G}_n)\right)^2 \quad (30)$$

$$= \mathbb{E}_{p_T}\left(\mathbf{1}(Y(X) = \hat{Y}(X)) - \alpha_T(\mathcal{G}_n; w^*) + \alpha_T(\mathcal{G}_n; w^*) - \hat{\alpha}_T(\mathcal{G}_n)\right)^2 \quad (31)$$

$$= \mathbb{E}_{p_T}\left[\left(\mathbf{1}(Y(X) = \hat{Y}(X)) - \alpha_T(\mathcal{G}_n; w^*)\right)^2 + (\alpha_T(\mathcal{G}_n; w^*) - \hat{\alpha}_T(\mathcal{G}_n))^2\right] \quad (32)$$

$$= \mathbb{E}_{p_T}\left(\mathbf{1}(Y(X) = \hat{Y}(X)) - \alpha_T(\mathcal{G}_n; w^*)\right)^2 + (\alpha_T(\mathcal{G}_n; w^*) - \hat{\alpha}_T(\mathcal{G}_n))^2 \quad (33)$$

where the equality (32) holds due to $\mathbb{E}_{T_X}\mathbb{E}_{T_{Y|X}}[(\mathbf{1}(Y(X) = \hat{Y}(X)) - \alpha_T(\mathcal{G}_n; w^*))(\alpha_T(\mathcal{G}_n; w^*) - \hat{\alpha}_T(\mathcal{G}_n))] = 0$. $\qquad\square$

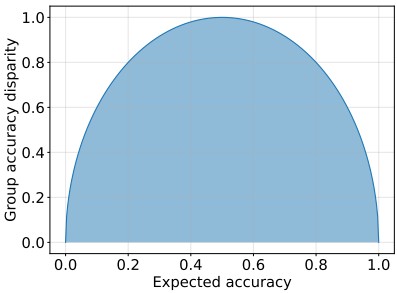

Figure A1: The shaded area includes values of group accuracy disparity satisfying the sufficient condition that the group accuracy estimator $\hat{\beta}^{(gr)}$ outperforms the individual group accuracy estimator $\hat{\beta}^{(id)}$.

## B Discussions

### B.1 Model calibration in the i.i.d. settings

In a classification problem, the maximum value of the softmax output is often considered as a confidence of a neural network's prediction. In Guo et al. (2017), it is shown that the modern neural networks are poorly calibrated, tending to produce larger confidences than their accuracies. Based on this observation, Guo et al. (2017) introduce a post-processing approach that adjusts a temperature parameter of the softmax function for adjusting the overall confidence level. In Bayesian approaches (such as Monte-Carlo dropout (Gal and Ghahramani, 2016; Gal et al., 2017), deep ensemble (Lakshminarayanan et al., 2017; Rahaman et al., 2021), and a last-layer Bayesian approach (Sensoy et al., 2018; Joo et al., 2020)), the confidence level adjustment is induced by posterior inference and model averaging. While both post-hoc calibration methods and Bayesian methods have been achieving impressive calibration performances in the i.i.d. setting (Maddox et al., 2019; Ovadia et al., 2019; Ebrahimi et al., 2020), it has been shown that most of the calibration improvement methods fall short under distribution shifts (Ovadia et al., 2019).

### B.2 On choice of non-parametric estimators

Our concept of determining the IW from its CI can be applied to any other valid CI estimators. For example, by analyzing a CI of the odds ratio of the logistic regression used as a domain classifier (Bickel et al., 2007; Park et al., 2020; Salvador et al., 2021), a CI of the IW can be obtained. Then, IW-GAE can be applied in the same way as developed in Section 4. As an extreme example, we apply IW-GAE by setting minimum and maximum values of IWs as CIs in an ablation study (Table A5). While IW-GAE outperforms strong baseline methods (CPCS and TransCal) even under this naive CI estimation, we observe that its performance is reduced compared to the setting with a sophisticated CI estimation discussed in Section 3. In this regard, advancements in IW estimation or CI estimation would be beneficial for accurately estimating the group accuracy, thereby model selection and uncertainty estimation. Therefore, we leave combining IW-GAE with advanced IW estimation techniques as an important future direction of research.

### B.3 On choice of the number of groups

In this work, we estimate the group accuracy by grouping predictions based on the confidence of the prediction. Therefore, a natural question to ask is how to select the number of groups. If we use a small number of groups, then there would be high $IdentBias(w^\dagger; \mathcal{G}_n)$ because of the large variance of prediction correctness within a group. In addition, reporting the same accuracy estimate for a large number of predictions could be inaccurate in terms of representing uncertainty for individual predictions. Conversely, if we use a large number of bins, there would be high Monte-Carlo approximation errors, $\epsilon_{stat}$. Therefore, it would result in a loose connection between the source group accuracy estimation error and the objective in the optimization problem (cf. Proposition 4.2). Therefore, it is important to choose a proper number of bins.

## C ADDITIONAL DETAILS

### C.1 A NESTED OPTIMIZATION PROBLEM UNDER THE TEMPERATURE SCALING IN THE TARGET DOMAIN

$$t^\dagger \in \underset{t \in \mathcal{T}}{\arg\min} \quad \sum_{n=1}^{M} \left( \hat{\alpha}_S^{(MC)}(\mathcal{G}_n) - \hat{\alpha}_S^{(IW)}(\mathcal{G}_n; w^\dagger(n; t)) \right)^2 \tag{34}$$

$$\text{where} \tag{35}$$

$$w^\dagger(n; t) \in \underset{\{w_i^{(S)}, w_i^{(T)}\}_{i \in [B]}}{\arg\min} \left( \hat{\alpha}_S^{(MC)}(\mathcal{G}_n) - \hat{\alpha}_S^{(IW)}(\mathcal{G}_n; \{w_i^{(S)}, w_i^{(T)}\}_{i \in [B]}) \right)^2 \tag{36}$$

$$\text{s.t.} \quad w_i^{(S)} \in \Phi_i, \quad \text{for } i \in [B] \tag{37}$$

$$w_i^{(T)} \in \Phi_i, \quad \text{for } i \in [B] \tag{38}$$

$$\| w_i^{(T)} - w_i^{(S)} \|_2^2 \leq \delta^{(tol)} \quad \text{for } i \in [B] \tag{39}$$

$$\left| \hat{\mathbb{E}}_{p_S}[\tilde{w}^{(S)}(X) | X \in \mathcal{G}_n] - \frac{\hat{P}(X_T \in \mathcal{G}_n^{(t)})}{\hat{P}(X_S \in \mathcal{G}_n)} \right| \leq \delta^{(prob)} \tag{40}$$

$$\left| \hat{\mathbb{E}}_{p_T}[1/\tilde{w}^{(T)}(X) | X \in \mathcal{G}_n^{(t)}] - \frac{\hat{P}(X_S \in \mathcal{G}_n)}{\hat{P}(X_T \in \mathcal{G}_n^{(t)})} \right| \leq \delta^{(prob)} \tag{41}$$

In the above optimization problem, we note that the temperature scaling also changes the IW based estimator by

$$\hat{\alpha}_S^{(IW)}(\mathcal{G}_n; \{w_i^{(S)}, w_i^{(T)}\}_{i \in [B]}) = \hat{\mathbb{E}}_{p_T} \left[ \frac{1}{\tilde{w}^{(T)}(X)} \middle| X \in \mathcal{G}_n^{(t)} \right] \hat{\mathbb{E}}_{p_S} \left[ \mathbf{1}(Y = \hat{Y})\tilde{w}^{(S)}(X) \middle| X \in \mathcal{G}_n \right]. \tag{42}$$

### C.2 ALGORITHM

In this section, we present pseudocodes of IW-GAE in Algorithm 1, evaluating calibration performance of IW-GAE in Algorithm 2, model selection by IW-GAE in Algorithm 3.

**Algorithm 1** Pseudocode of IW-GAE

**Input:** Source dataset $\mathcal{D}_S = \{(x_i^{(S)}, y_i^{(S)})\}_{i=1}^{N^{(S)}}$, Target dataset $\mathcal{D}_T = \{x_i^{(T)}\}_{i=1}^{N^{(T)}}$
**Hyperparameters:** The numbers of bins and groups ($B$ and $M$) level of CI $\delta$, search space $\mathcal{T}$
*# Prepare a UDA model (Wang et al., 2020)*
Partition $\mathcal{D}_S$ into $\mathcal{D}_S^{tr}$ and $\mathcal{D}_S^{val}$
Train a neural network $g$ on $(\mathcal{D}_S^{tr}, \mathcal{D}_T)$ with any UDA method
Upsample $\mathcal{D}_S^{tr}$ or $\mathcal{D}_T$ to make $|\mathcal{D}_S^{tr}| = |\mathcal{D}_T|$
Compute $\mathcal{F}_S^{tr} = \{g(x)|x \in \mathcal{D}_S^{tr}\}$, $\mathcal{F}_S^{val} = \{g(x)|x \in \mathcal{D}_S^{val}\}$, and $\mathcal{F}_T = \{g(x)|x \in \mathcal{D}_T\}$
Train a logistic regression model $H$ that discriminates $\mathcal{F}_S^{tr}$ and $\mathcal{F}_T$
*# Obtain CIs of IWs (Park et al., 2022)*
Gather IWs $\mathcal{W}_{S \cup T} = \{(1 - H(g(x)))/H(g(x)) : x \in \mathcal{D}_S^{val} \cup \mathcal{D}_T\}$
Compute quantiles $q(i) = i/(B+1)$-th quantile of $\mathcal{W}_{S \cup T}$ for $i \in [B+1]$
Construct bins $\mathcal{B}_i = \{x \in \mathcal{D}_S^{val} \cup \mathcal{D}_T : q(i) \leq (1 - H(g(x)))/H(g(x)) \leq q(i+1)\}$ for $i \in [B]$
Compute $\Phi_i$ using (2) for each $i \in [B]$
*# IW-GAE*
$f^\dagger = \infty$
**for** $t \in \mathcal{T}$ **do**
    Obtain $w(n; t)$ by solving the optimization problem in (34)-(41) for $n \in [M]$
    **if** $\sum_{n=1}^{M} \left( \hat{\alpha}_S^{(MC)}(\mathcal{G}_n) - \hat{\alpha}_S^{(IW)}(\mathcal{G}_n; w(n; t)) \right)^2 \leq f^\dagger$ **then**
        $f^\dagger = \sum_{n=1}^{M} \left( \hat{\alpha}_S^{(MC)}(\mathcal{G}_n) - \hat{\alpha}_S^{(IW)}(\mathcal{G}_n; w(n; t)) \right)^2$
        $t^\dagger = t$
        $w^\dagger(n) = w(n; t)$ for $n \in [M]$
    **end if**
**end for**
**return** $(t^\dagger, w^\dagger(n))$

---

**Algorithm 2** Pseudocode of evaluating calibration performance of IW-GAE

**Input:** Labeled test dataset on the target domain $\mathcal{D}_T^{test} = \{(x_i^{(T,*)}, y_i^{(T,*)})\}_{i=1}^{N^{(T,*)}}$, target group accuracy estimators of IW-GAE $\{\hat{\alpha}_T(\mathcal{G}_i; w^\dagger(i))\}_{i=1}^{M}$, an optimal temperature $t^\dagger$, a neural network logit function $g$
$ECE(\mathcal{D}_T^{test}) \leftarrow 0$
Define a group ID function $ID(x) = k$ if $\frac{k}{M} \leq \phi(g(x)/t^\dagger) \leq \frac{k+1}{M}$
*# ECE Measures*
**for** $m \in [M]$ **do**
    *# The prediction confidence at $x$ is computed by the estimated accuracy of the group that contains $x$, not the maximum value of the softmax output at $x$*
    Gather samples in $m$-th confidence group $\mathcal{G} = \{x \in \mathcal{D}_T^{test} : \frac{m}{M} \leq \hat{\alpha}_T(\mathcal{G}_{ID(x)}; w^\dagger(ID(x))) \leq \frac{m+1}{M}\}$
    Compute average confidence $\hat{\text{Conf}} = \frac{1}{|\mathcal{G}|} \sum_{x \in \mathcal{G}} \hat{\alpha}_T(\mathcal{G}_{ID(x)}; w^\dagger(ID(x)))$
    Compute average accuracy of $\mathcal{G}$, and denote it as $\hat{\text{Acc}}$
    Compute calibration error of $m$-th group confidence $v = |\hat{\text{Acc}} - \hat{\text{Conf}}|$
    $ECE(\mathcal{D}_T^{test}) += \frac{|\mathcal{G}|}{|\mathcal{D}_T^{test}|} \cdot v$
**end for**
**return** $ECE(\mathcal{D}_T^{test})$

---

**Algorithm 3** Pseudocode of model selection by IW-GAE

---

**Input:** Unlabeled dataset on the target domain $\mathcal{D}_T = \{x_i^T\}_{i=1}^{N^T}$, target group accuracy estimators of IW-GAE $\{\hat{\alpha}_T(\mathcal{G}_i; w^\dagger(i))\}_{i=1}^{M}$, an optimal temperature $t^\dagger$, a set of neural network logit functions $\{g_i\}_{i=1}^{Z}$
Initialize $f^\dagger = 0$
**for** $z \in [Z]$ **do**
    Define a group ID function $ID(x) = k$ if $\frac{k}{M} \leq \phi(g_z(x)/t^\dagger) \leq \frac{k+1}{M}$
    # *Evaluate model*
    Compute the average target group accuracy $f = \frac{1}{N^T} \sum_{i=1}^{N^T} \hat{\alpha}_T(\mathcal{G}_{ID(x_i^T)}; w^\dagger(ID(x_i^T)))$
    **if** $f \geq f^\dagger$ **then**
        $f^\dagger = f$
        $z^* = z$
    **end if**
**end for**
**return** $g_{z^*}$

---

# D EXPERIMENTAL DETAILS

We follow the exact same training configurations as those used in the Transfer Learning Library, except we separate 20% as the validation dataset from the source domain (in the original implementation, validation is performed with the test dataset for Office-Home).

The configuration of training MDD for Office-Home is as follows: MDD is trained for 30 epochs with SGD with momentum parameter 0.9 and weight decay of $0.0005$. The learning rate is schedule by $\alpha \cdot (1 + \gamma \cdot t)^{-\eta}$ where $t$ is the iteration counter, $\alpha = 0.004$, $\gamma = 0.0002$, $\eta = 0.75$, and the stochastic gradient is computed with minibatch of 32 samples from the source domain and 32 samples from the target domain. Also, it uses the margin coefficient of 4 as the MDD-specific hyperparamter. For the model architecture, it uses ResNet-50 pre-trained on ImageNet (Russakovsky et al., 2015) with the bottleneck dimension of 2,048.

**IW-GAE specific details** For CI estimation, we follow the same configuration with the original method (Park et al., 2022). Specifically, we use constant $G = 0.001$, CI level $\bar{\delta} = 0.05$, and the number of bins $B = 10$. In addition, we use the maximum IW value $\tilde{w}_n^{(ub)} = 6.0$ and the minimum IW value $\underline{w}^\dagger(n) = 1/6$ for $n \in [M]$ for tightening the upper bounds in (5) and (15) (cf. Section 4.2).

In addition, for IW-GAE, we use the constraint relaxation constants $\delta^{(tol)} = 0.1$ and $\delta^{(prob)} = 0.3$. We also use the number of accuracy group $M = 10$ and analyze the sensitivity for $M$ in Appendix E.6. For implementing sequential least square programming, we use the SciPy Library (Virtanen et al., 2020) with tolerance $10^{-8}$ that is used to check a convergence condition (other optimizer-specific values follow the default values in SciPy) and choose the middle points from CIs of binned IW as an initial solution. For nested optimization, we set $\mathcal{T} := \{0.85, 0.90, 0.95, 1.00, 1.05, 1.10\}$.

# E ADDITIONAL EXPERIMENTS

## E.1 LARGE-SCALE IMAGE CLASSIFICATION TASK

We perform an additional large-scale experiment with the VisDA-2017 (Peng et al., 2017) containing around 280,000 images of 12 categories from two domains (real and synthetic images). For this experiment, we use ResNet-101 with the bottleneck dimension of 1,024, following the default setting in the Transfer Learning Library. All other configurations are the same as the training configuration of the Office-Home dataset (cf. Appendix D). Figure A2 represents the result of the VisDa-2017 experiment, which shows that the IW-GAE achieves the best performance among all baselines. Specifically, IW-GAE reduces the ECE of the state-of-the-art (TransCal) by 21%. We note that the IW-Mid achieves a comparable performance with TransCal, unlike the OfficeHome experiment. This result could be explained by an accurate CI estimation under a large number of samples. We leave analyzing the impacts of the number of samples on the CI estimation and the performance of IW-GAE

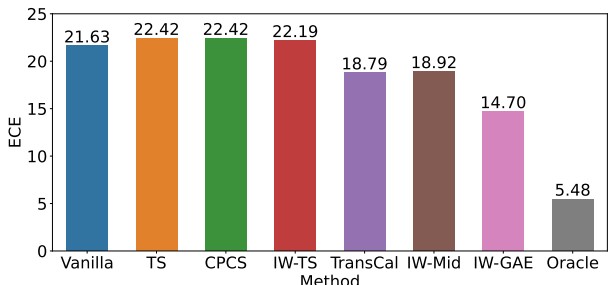

Figure A2: Large-scale model calibration benchmark result of MDD with ResNet-101 on VisDA 2017. The numbers indicates the mean ECE across ten repetitions with boldface for the minimum mean ECE.

| Method | Cl-Pt | Cl-Rw | Cl-Sk | Pt-Cl | Pt-Rw | Pt-Sk | Rw-Cl | Rw-Pt | Rw-Sk | Sk-Cl | Sk-Pt | Sk-Rw | Avg |
|--------|-------|-------|-------|-------|-------|-------|-------|-------|-------|-------|-------|-------|-----|
| Vanilla | 13.23 | 6.36 | 12.92 | 9.75 | 6.35 | 15.56 | 9.44 | 9.70 | 14.34 | 6.63 | 11.25 | **5.23** | 10.06 |
| TS | 12.95 | **5.95** | 13.32 | 6.40 | 3.90 | 11.07 | 8.64 | 10.49 | 16.08 | **3.17** | 5.58 | 13.09 | 9.22 |
| CPCS | **5.64** | 21.90 | 7.70 | **5.14** | 7.72 | 7.90 | 9.35 | 11.17 | 17.06 | 3.46 | **2.23** | 15.90 | 9.60 |
| IW-TS | 16.76 | 16.7 | 12.53 | 5.29 | 7.84 | 4.34 | 9.60 | 10.58 | 16.80 | 5.40 | 2.98 | 17.11 | 10.49 |
| TransCal | 18.51 | 29.63 | 20.92 | 23.02 | 31.83 | 17.58 | 27.88 | 28.83 | 20.31 | 31.66 | 23.06 | 31.46 | 25.39 |
| IW-Mid | 7.61 | 11.01 | 5.89 | 8.84 | 7.58 | 5.36 | 8.70 | 7.49 | 7.53 | 10.24 | 8.10 | 10.21 | 8.21 |
| IW-GAE | 6.06 | 8.15 | **5.38** | 7.45 | **3.89** | **3.94** | **7.01** | **5.58** | **6.73** | 6.80 | 6.82 | 8.00 | **6.32** |
| Oracle | 4.55 | 2.78 | 4.01 | 3.10 | 3.72 | 2.72 | 3.10 | 2.79 | 2.83 | 3.13 | 1.70 | 1.77 | 3.02 |
| TransCal- | 6.36 | 25.28 | 9.71 | 0.23 | 7.20 | 0.20 | 37.93 | 13.33 | 11.7 | 4.12 | 11.59 | 24.09 | 12.65 |

Table A1: Large-scale model calibration benchmark results of CDAN with ResNet-50 on DomainNet. The numbers indicates the mean ECE across ten repetitions with boldface for the minimum mean ECE. Cl, Pt, Rw, and Sk correspond to clipart, painting, real, and sketch, respectively.

as an important future direction of research, which could provide a strategy for choosing which CI estimator to use under which condition.

To enhance the robustness of the findings, we also perform another large-scale experiment with DomainNet (Peng et al., 2019) containing around 570,000 images of 345 categories from 6 domains (clipart, real, sketch, infograph, painting, quickdraw). We also follow the default setting in the Transfer Learning Library, except replacing ResNet-101 by ResNet-50 due to the computational budget. Also, we use only four domains (clipart, real, sketch, painting) as described in the Transfer Learning Library. Table A1 presents the result of the DomainNet experiment, which shows that IW-GAE achieves the best performance among all baselines, achieving 31% lower ECE than the second best method (TS). Consistent to the finding in VisDA-2017 (Peng et al., 2017), IW-Mid achieves an impressive performance, surpassing all baseline methods on average. Finally, we remark that TransCal's performance on DomainNet is significantly worse than its performance on other datasets. We found that excluding a variance reduction term in TransCal (TransCal- in Table A1) stabilizes its performance, but the performances are still lower than IW-GAE and other baselines (TS and CPCS).

## E.2 EXPERIMENTS WITH DIFFERENT BASE MODELS

In this section, we show the effectiveness of IW-GAE with two different base models. First, we perform additional experiments with conditional domain adversarial network (CDAN; (Long et al., 2018)) which is also a popular UDA method. As in the experiments with MDD, we use ResNet-50 as the backbone network and OfficeHome as the dataset. The learning rate schedule for CDAN is $\alpha \cdot (1 + \gamma \cdot t)^{-\eta}$ where $t$ is the iteration counter, $\alpha = 0.01$, $\gamma = 0.001$, and $\eta = 0.75$. The remaining training configuration for CDAN is the same as the MDD training configuration except it uses the bottleneck dimension of 256 and weight decay of 0.0005 (cf. Appendix D). As we can see from Table A2, IW-GAE achieves the best performance among all considered methods, achieving the best ECE in 8 out of the 12 cases as well as the lowest mean ECE. We note that TransCal achieves a performance comparable to IW-GAE in this experiment, but considering the results in the other tasks, IW-GAE is still an appealing method for performing the model calibration task.

| Method | Ar-Cl | Ar-Pr | Ar-Rw | Cl-Ar | Cl-Pr | Cl-Rw | Pr-Ar | Pr-Cl | Pr-Rw | Rw-Ar | Rw-Cl | Rw-Pr | Avg |
|---|---|---|---|---|---|---|---|---|---|---|---|---|---|
| Vanilla | 30.73 | 18.38 | 14.37 | 25.63 | 22.44 | 19.10 | 27.54 | 36.72 | 12.48 | 19.93 | 31.12 | 10.88 | 22.44 |
| TS | 29.68 | 19.40 | 14.40 | 22.15 | 19.97 | 16.88 | 28.82 | 38.03 | 12.99 | 20.46 | 31.91 | 11.83 | 22.21 |
| CPCS | 18.78 | 18.09 | 14.74 | 22.18 | 20.74 | 16.33 | 29.30 | 34.92 | 11.92 | 20.99 | 31.41 | 11.07 | 20.87 |
| IW-TS | 12.38 | 16.79 | 14.85 | 21.75 | 20.06 | 16.92 | 29.30 | 38.84 | 13.30 | 20.82 | 31.10 | 11.37 | 20.62 |
| TransCal | **7.94** | **14.05** | 12.91 | 7.82 | 9.25 | 10.23 | 9.37 | **12.60** | 14.29 | **9.92** | 17.51 | 11.30 | |
| IW-Mid | 36.05 | 47.70 | 26.82 | 21.08 | 22.95 | 21.55 | 18.88 | 28.99 | 15.39 | 21.16 | 28.16 | 25.27 | 26.17 |
| IW-GAE | 13.98 | 29.82 | **9.44** | **6.55** | **5.59** | **10.16** | **5.29** | 13.47 | **11.01** | 11.12 | **7.26** | **9.84** | **11.13** |
| Oracle | 7.91 | 8.80 | 6.05 | 7.57 | 7.93 | 6.76 | 9.07 | 9.14 | 4.04 | 7.16 | 9.19 | 5.65 | 7.44 |

Table A2: Model calibration benchmark results of CDAN with ResNet-50 on Office-Home. The numbers indicates the mean ECE across ten repetitions with boldface for the minimum mean ECE.

| Method | Ar-Cl | Ar-Pr | Ar-Rw | Cl-Ar | Cl-Pr | Cl-Rw | Pr-Ar | Pr-Cl | Pr-Rw | Rw-Ar | Rw-Cl | Rw-Pr | Avg |
|---|---|---|---|---|---|---|---|---|---|---|---|---|---|
| Vanilla | 38.91 | 26.39 | 18.86 | 32.85 | 26.69 | 19.36 | 35.87 | 36.70 | 18.61 | 24.57 | 36.87 | 14.79 | 27.54 |
| TS | 31.84 | 22.55 | 13.49 | 26.16 | 20.10 | 10.72 | 33.98 | 31.91 | 15.59 | 21.62 | 31.59 | 12.46 | 22.67 |
| CPCS | 13.07 | 20.09 | 47.15 | **9.78** | 21.82 | 8.02 | 32.65 | 25.61 | 15.27 | 20.53 | 40.38 | 7.84 | 21.85 |
| IW-TS | **12.88** | 21.44 | 61.15 | 10.56 | 16.40 | 11.72 | 33.03 | 36.37 | 14.09 | 19.96 | 41.95 | 19.30 | 24.91 |
| TransCal | 19.23 | 15.09 | 6.55 | 17.91 | 11.60 | 3.91 | **22.98** | **15.81** | **6.11** | 13.77 | 21.40 | 4.02 | 13.2 |
| IW-Mid | 50.68 | 28.93 | 23.92 | 38.24 | 33.48 | 28.58 | 39.76 | 37.45 | 22.40 | 27.15 | 44.15 | 18.07 | 32.73 |
| IW-GAE | 22.21 | **10.68** | **2.38** | 15.96 | **9.30** | **3.53** | 23.54 | 22.73 | 6.37 | **11.78** | **20.75** | **1.63** | **12.57** |
| Oracle | 5.88 | 9.91 | 3.19 | 7.75 | 4.64 | 3.66 | 4.17 | 7.70 | 3.09 | 4.51 | 8.09 | 3.54 | 5.51 |

Table A3: Model calibration benchmark results of MCD with ResNet-50 on Office-Home. The numbers indicates the mean ECE across ten repetitions with boldface for the minimum mean ECE.

We also perform additional experiments with maximum classifier discrepancy (MCD; (Saito et al., 2018)). Following the previous experiments with MDD and CDAN, we use ResNet-50 as the backbone network and OfficeHome as the dataset. The training configuration is the same as the MDD training configuration except it uses the fixed learning rate of 0.001 with weight decay of 0.0005 and bottleneck dimension of 1,024 (cf. Appendix D). Consistent to other benchmark results, IW-GAE achieves the best performance among all methods (Table A3). Specifically, IW-GAE achieves the best average model calibration performance, and its ECE is lowest in 7 out of 12 domains. Note that IW-Mid's performance with MCD is significantly lower compared to other benchmark results. However, IW-GAE still significantly improves the performance, indicating that IW-GAE does not strongly depends on accuracy of the CI estimation discussed in Section 3.

### E.3 CHECKPOINT SELECTION

In this section, we perform the task of choosing the best checkpoint during training for examining IW-GAE's model selection performance. Specifically, we first train MDD on the OfficeHome dataset for 30 epochs and save the checkpoint at the end of each epoch. Then, we choose the best checkpoint based on IW-GAE and baselines described in Section 5.2. As shown in Table A4, the model selected based on IW-GAE achieves the best average test accuracy, which is consistent with the results in the hyperparameter selection task (cf. the model selection task in Table 1). Specifically, IW-GAE improves the second-best method (IWCV) by 9% and achieves the best checkpoint selection for 3 out of the 12 domains.

| Method | Ar-Cl | Ar-Pr | Ar-Rw | Cl-Ar | Cl-Pr | Cl-Rw | Pr-Ar | Pr-Cl | Pr-Rw | Rw-Ar | Rw-Cl | Rw-Pr | Avg |
|---|---|---|---|---|---|---|---|---|---|---|---|---|---|
| Vanilla | 47.22 | 74.14 | 77.76 | 61.85 | 70.96 | 71.59 | 60.98 | 53.63 | **78.93** | 71.57 | 57.04 | 83.96 | 67.47 |
| IWCV | **54.46** | **74.22** | 72.27 | 61.48 | 70.49 | 70.62 | 61.30 | 51.13 | 78.37 | 72.94 | 58.43 | 84.00 | 67.48 |
| DEV | 54.04 | 73.94 | 78.16 | 61.52 | 63.19 | 70.7 | 60.43 | 53.63 | 78.93 | 71.57 | 58.62 | 83.89 | 67.39 |
| IW-Mid | 54.04 | 72.63 | 78.37 | **62.05** | **71.28** | 71.45 | **61.25** | **54.39** | **79.07** | 73.19 | **58.75** | 80.06 | 68.04 |
| IW-GAE | 54.32 | 73.98 | **78.51** | 61.96 | 71.25 | **71.70** | 61.10 | 54.30 | 78.91 | **73.22** | 58.70 | **83.86** | **68.48** |
| Lower bound | 41.90 | 64.88 | 72.27 | 52.00 | 58.48 | 62.13 | 53.52 | 38.33 | 70.92 | 63.41 | 44.81 | 75.83 | 58.21 |
| Oracle | 54.80 | 74.79 | 78.61 | 62.46 | 71.59 | 72.18 | 61.64 | 54.64 | 79.44 | 73.42 | 59.43 | 84.12 | 68.93 |

Table A4: Checkpoint selection benchmark results of MDD with ResNet-50 on Office-Home. The numbers indicate the mean test accuracy of selected model across ten repetitions with boldface for the maximum mean test accuracy.

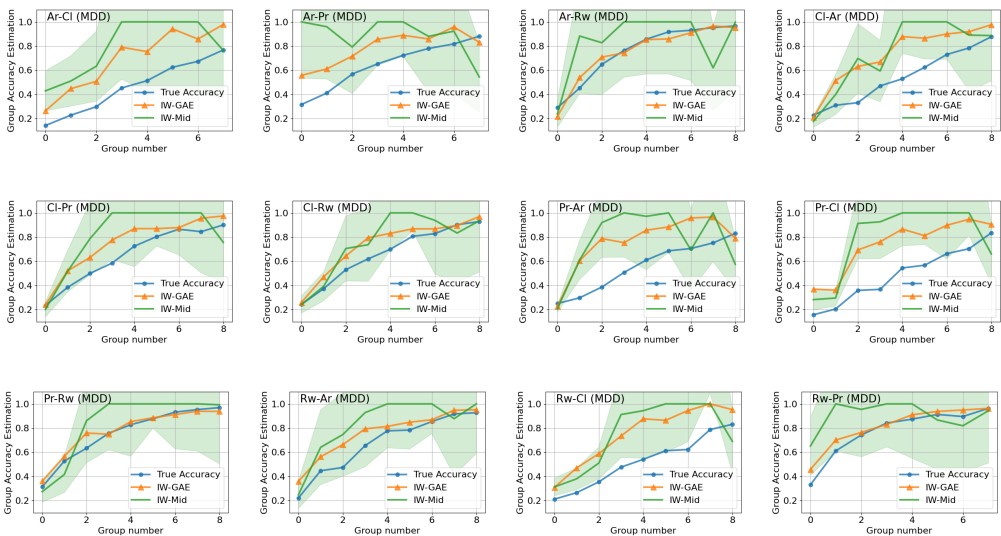

Figure A3: True group accuracy and estimated group accuracy of IW-GAE and IW-Mid under MDD. The shaded areas represent possible group accuracy estimation with binned IWs in the CI. The title of a figure represents "Source-Target." For IW-Mid and IW-GAE, we clip the accuracy estimations when they exceed 1, which can occur when the upper bound of CI is large. Also, the number of groups in the figure is different for some domains because there can be a group that contains no target samples (we set $M = 10$ for all cases).

### E.4 QUALITATIVE EVALUATION OF IW-GAE

To qualitatively analyze IW-GAE, we also visualize reliability curves that compare the estimated group accuracy with the average accuracy in Figure A3. We first note that IW-GAE tends to accurately estimate the true group accuracy for most groups under different cases compared to IW-Mid. The accurate group accuracy estimation behavior of IW-GAE explains the results that the IW-GAE improves IW-Mid for most cases in the model calibration and selection tasks (cf. Table 1). For most cases, true accuracy is in between the lower and upper IW estimators, albeit the interval length tends to increase for high-confidence groups. This means that the CI of the IW based on the Clopper-Pearson method successfully captures the IW in the CI. We also note that the true accuracy is close to the lower IW estimator in the lower confidence group and the middle IW estimator in the high confidence group. An observation that the true accuracy's relative positions in CIs varies from one group to another group motivates why an adaptive selection of binned IWs as ours is needed.

### E.5 ABLATION STUDY

In this section, we perform an ablation study of our key design choices for group construction (cf. Section 4.2). The first ablation study examines group construction based on the maximum value of the softmax output by constructing a group function based on IW. The second ablation study examines the effectiveness of our nested optimization with temperature scaling (cf. Section 4.2), by excluding the outer optimization in the nested optimization; i.e., setting $\mathcal{T} = \{1\}$ in Appendix C.1. Specifically, we repeat the model calibration experiment with MDD on four randomly selected domains in the OfficeHome dataset (Ar-Pr, Pr-Cl, Rw-Cl, Rw-Pr).

Table A5 presents the results of the ablation study. First, note that the grouping by the maximum value of softmax significantly impacts the performance of IW-GAE. If we assume the group accuracy estimation ability of IW-GAE is not significantly reduced after changing the grouping function $I^{(g)}$, the reduction in the performance could be due to a large variance of prediction accuracy within a group (cf. the case of group 2 in Figure 1(b)). Specifically, the large value of $Var(\mathbf{1}(Y = \hat{Y})|\mathcal{G}_n)$ increases the $IdentBias(w^\dagger(n); \mathcal{G}_n)$, which can loosen the upper bound of the source group accuracy estimation error in (15). Given its significant impact on the calibration performance, we want to

| Method | Ar-Pr | Pr-Cl | Rw-Cl | Rw-Pr | Avg |
|---|---|---|---|---|---|
| Vanilla | 40.61 | 38.62 | 36.51 | 14.01 | 32.44 |
| CPCS | 22.07 | 29.20 | 26.54 | 11.14 | 22.24 |
| TransCal | 20.27 | 29.86 | 29.90 | 10.00 | 22.51 |
| IW-GAE w/ grouping by IW and w/o the nested optimization | 15.07 | 36.02 | 35.01 | 5.23 | 22.83 |
| IW-GAE w/ grouping by IW | 14.18 | 34.67 | 35.08 | 5.30 | 22.31 |
| IW-GAE w/o the nested optimization | 11.43 | 16.57 | 9.38 | 5.91 | 10.82 |
| IW-GAE w/o the CI estimation (Park et al., 2022) | 11.00 | 29.73 | 24.44 | 2.09 | 16.82 |
| IW-GAE | 4.70 | 17.49 | 9.52 | 8.14 | 9.97 |

Table A5: An ablation study of key design choices of constructing a group by the maximum value of the softmax and using nested optimization with temperature scaling on the target domain for IW-GAE. "IW-GAE w/o the CI estimation (Park et al., 2022)" corresponds to the setting where $\Phi_i = [1/6, 6.0]$ for $i \in [M]$. We use MDD with ResNet-50 on Office-Home. The numbers indicate the mean ECE across ten repetitions.

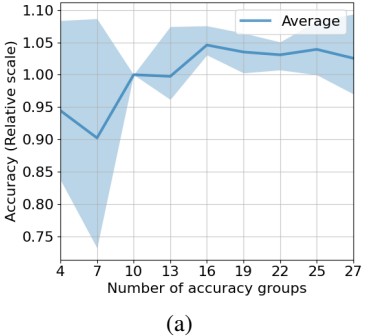

(a)

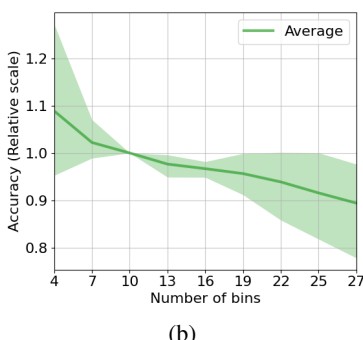

(b)

Figure A4: Sensitivity analysis of IW-GAE with respect to the number of groups $M$ and the number of bins $B$ (b) on four domains (Ar-Pr, Pr-Cl, Rw-Cl, Rw-Pr) in the OfficeHome dataset. Default hyperparameters for $M$ and $B$ are 10, and we normalize the accuracy for each domain by its performance under the default hyperparameters. Average represents the average relative accuracy for each value of the hyperparameter and the shaded areas represent areas between the minimum and the maximum relative accuracy over the 4 domains.

remark few challenging aspects of developing an ideal group function for future work. Specifically, note that the core factor in $IdentBias(w^\dagger(n); \mathcal{G}_n)$ impacted by $I^{(g)}$ is $Var(\mathbf{1}(Y = \hat{Y})|\mathcal{G}_n)$. This term depends on the labeled information in the target domain, so it is hard to foretell changes in $IdentBias(w^\dagger(n); \mathcal{G}_n)$ as we change $I^{(g)}$. Furthermore, even if we have a labeled dataset in the target domain, finding the optimal $I^{(g)}$ that minimizes $IdentBias(w^\dagger(n); \mathcal{G}_n)$ is a combinatorial optimization problem, which is one of the most challenging optimization problems.

Next, note that nested optimization results in improvement of the ECE on average but it is effective only in four out of eight cases. This is surprising because the nested optimization problem is guaranteed to reduce $\epsilon_{opt}(w^\dagger(n))$ as $\mathcal{T}$ contains the case that corresponds to the setting without the nested optimization; $1 \in \mathcal{T}$. This motivates our further investigation of the relationship between $\epsilon_{opt}(w^\dagger(n))$ and $IdentBias(w^\dagger(n); \mathcal{G}_n)$ in Appendix E.7. In a nutshell, we found the cases when reducing $\epsilon_{opt}(w^\dagger(n))$ increases $IdentBias(w^\dagger(n); \mathcal{G}_n)$. Therefore, it can increase the upper bound of a source group accuracy estimation error in (15). We also found a (weak) correlation between $\epsilon_{opt}(w^\dagger(n))$ and $IdentBias(w^\dagger(n); \mathcal{G}_n)$, which can explain the average improvement by the nested optimization problem. We present more details in Appendix E.7.

### E.6 SENSITIVITY ANALYSIS

In this section, we conduct a sensitivity analysis with respect to our key hyperparameters of the number of accuracy groups $M$ and the number of bins $B$. As in the ablation study in Appendix E.5, we perform the model calibration experiment with MDD on the four domains in the OfficeHome dataset

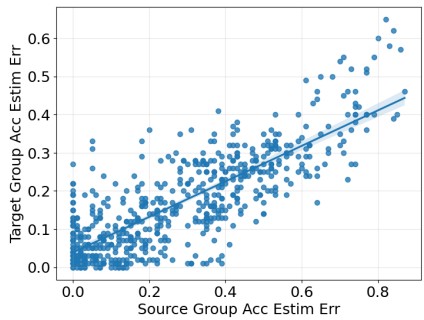

Figure A5: The relationship between source and target group accuracy estimation errors.

(Ar-Pr, Pr-Cl, Rw-Cl, Rw-Pr). Figure A4 shows the sensitivity analysis results with hyperparameter values $M \in [4, 27]$ and $B \in [4, 27]$ (the default value for both $M$ and $B$ is 10). Within the search space, the average performances do not change more than 10%, which means that IW-GAE would outperform state-of-the-art (cf. Table 1) also under such altered settings. Also, we can see that the performance changes under different hyperparameter values are somewhat stable; the maximum and minimum changes are within the range of 10% for most cases, even though a large variance in the performances appears for extreme values such as $M = 4$, $B = 4$, and $B = 27$. The results show the robustness of IW-GAE with respect to small changes in the key hyperparameter values.

### E.7 ANALYSIS OF $\epsilon_{opt}(w^\dagger(n))$ AND $IdentBias(w^\dagger(n); \mathcal{G}_n)$ OF IW AND THEIR RELATION TO SOURCE AND TARGET GROUP ACCURACY ESTIMATION ERRORS

In this section, we aim to answer the following question about the central idea of this work: *"Does solving the optimization problem in (8)-(13) result in an accurate target group accuracy estimator?"* Specifically, we analyze the relationship between the optimization error $\epsilon_{opt}(w^\dagger(n))$, the bias of the identical accuracy assumption $IdentBias(w^\dagger(n); \mathcal{G}_n)$, the source group accuracy estimation error $|\alpha_S(\mathcal{G}_n; w^*) - \alpha_S(\mathcal{G}_n; w^\dagger(n))|$, and the target group accuracy estimation error $|\alpha_T(\mathcal{G}_n; w^*) - \alpha_T(\mathcal{G}_n; w^\dagger(n))|$ from the perspective of (5) and (15)[2]. To this end, we gather $w^\dagger(n)$ obtained by solving the optimization problem under all temperature parameters in the search space $t \in \mathcal{T}$ with MDD on the OfficeHome dataset (720 IWs from 6 values of the temperature parameter, 12 cases, and 10 groups). Then, by using the test dataset in the source and the target domains, we obtain the following observations.

In (5), we show that $|\alpha_T(\mathcal{G}_n; w^*) - \alpha_T(\mathcal{G}_n; w^\dagger(n))|$ is upper bounded by $|\alpha_S(\mathcal{G}_n; w^*) - \alpha_S(\mathcal{G}_n; w^\dagger(n))|$. However, the inequality could be loose since the inequality is obtained by taking the maximum over the IW values. Considering that the optimization problem is formulated for finding $w^\dagger(n)$ that achieves small $|\alpha_S(\mathcal{G}_n; w^*) - \alpha_S(\mathcal{G}_n; w^\dagger(n))|$ (cf. Proposition 4.2), the loose connection between the source and target group accuracy estimation errors can potentially enlighten a fundamental difficulty to our approach. However, as we can see from Figure A5, it turns out that $|\alpha_S(\mathcal{G}_n; w^*) - \alpha_S(\mathcal{G}_n; w^\dagger(n))|$ is strongly correlated with $|\alpha_T(\mathcal{G}_n; w^*) - \alpha_T(\mathcal{G}_n; w^\dagger(n))|$. This result validates our approach of reducing the source accuracy estimation error of the IW-based estimator for obtaining an accurate group accuracy estimator in the target domain.

In (15), we show that $|\alpha_S(\mathcal{G}_n; w^*) - \alpha_S(\mathcal{G}_n; w^\dagger(n))| \leq \epsilon_{opt}(w^\dagger(n)) + \epsilon_{stat} + IdentBias(w^\dagger(n); \mathcal{G}_n)$, which motivates us to solve the optimization problem for reducing $\epsilon_{opt}(w^\dagger(n))$ (cf. Section 4.2) and to construct groups based on the maximum value of softmax for reducing $IdentBias(w^\dagger(n); \mathcal{G}_n)$ (cf. Section 4.2). Again, if these terms are loosely connected to $|\alpha_S(\mathcal{G}_n; w^*) - \alpha_S(\mathcal{G}_n; w^\dagger(n))|$, a fundamental difficulty arises for our approach. In this regard, we analyze the relationship between $\epsilon_{opt}(w^\dagger(n))$, $IdentBias(w^\dagger(n); \mathcal{G}_n)$, and $|\alpha_S(\mathcal{G}_n; w^*) - \alpha_S(\mathcal{G}_n; w^\dagger(n))|$. From Figure A6, we can see that both $\epsilon_{opt}(w^\dagger(n))$ and $IdentBias(w^\dagger(n); \mathcal{G}_n)$ are strongly correlated to the source group accuracy estimation error. Combined with the observation in

---

[2]Technically speaking, the computed values in this experiment are the empirical expectation which can contain a statistical error. However, since we have no access to the data generating distribution, we perform the analysis as if these values are the population expectations.

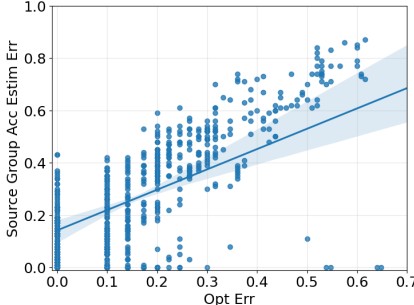 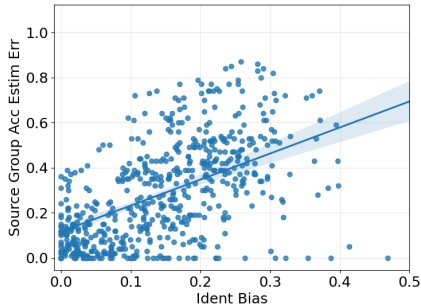

Figure A6: The relationship between between $\epsilon_{opt}(w^{\dagger}(n))$ and the source group accuracy estimation error (left) and the relationship between $IdentBias(w^{\dagger}(n); \mathcal{G}_n)$ and the source group accuracy estimation error (right).

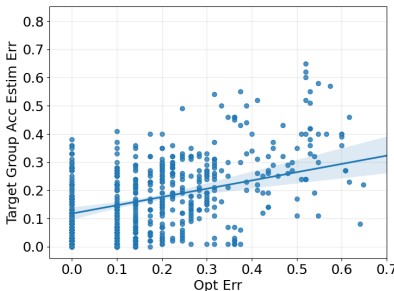

Figure A7: The relationship between $\epsilon_{opt}(w^{\dagger}(n))$ and the target group accuracy estimation error.

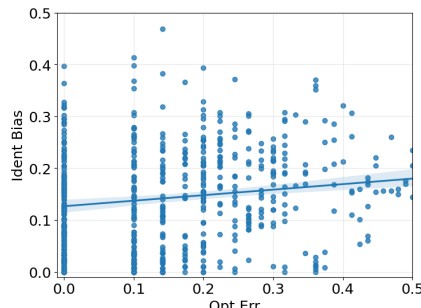

Figure A8: The relationship between $\epsilon_{opt}(w^{\dagger}(n))$ and $IdentBias(w^{\dagger}(n); \mathcal{G}_n)$.

Figure A5, this observation explains the impressive performance gains by IW-GAE developed for reducing $\epsilon_{opt}(w^{\dagger}(n))$ and $IdentBias(w^{\dagger}(n); \mathcal{G}_n)$.

Next, we analyze the efficacy of solving the optimization problem for obtaining an accurate target group accuracy estimator. To this end, we analyze the relationship between $\epsilon_{opt}(w^{\dagger}(n))$ and $|\alpha_T(\mathcal{G}_n; w^*) - \alpha_T(\mathcal{G}_n; w^{\dagger}(n))|$. From Figure A7, $\epsilon_{opt}(w^{\dagger}(n))$ is correlated with $|\alpha_T(\mathcal{G}_n; w^*) - \alpha_T(\mathcal{G}_n; w^{\dagger}(n))|$, which explains the performance gains in the model calibration and selection tasks by IW-GAE. However, the correlation is weaker than the cases analyzed in Figure A5 and Figure A6. We conjecture that this is because $\epsilon_{opt}(w^{\dagger}(n))$ is connected to $|\alpha_T(\mathcal{G}_n; w^*) - \alpha_T(\mathcal{G}_n; w^{\dagger}(n))|$ through two inequalities (5) and (15), and this results in a somewhat loose connection between $\epsilon_{opt}(w^{\dagger}(n))$ and $|\alpha_T(\mathcal{G}_n; w^*) - \alpha_T(\mathcal{G}_n; w^{\dagger}(n))|$.

In Figure A7, we also note that the optimization problem is subject to a non-identifiability issue that the solutions with the same optimization error can have significantly different target group accuracy estimation errors (e.g., points achieving the zero optimization error in Figure A7). We remark that the non-identifiability issue motivates an important future direction of research that develops a more sophisticated objective function and a regularization function that can distinguish estimators with different target group accuracy estimation errors.

Finally, we analyze the relationship between $\epsilon_{opt}(w^{\dagger}(n))$ and $IdentBias(w^{\dagger}(n); \mathcal{G}_n)$ in Figure A8. In general, we can see a weak correlation between $\epsilon_{opt}(w^{\dagger}(n))$ and $IdentBias(w^{\dagger}(n); \mathcal{G}_n)$. This means that, in general, finding a better solution in terms of $\epsilon_{opt}(w^{\dagger}(n))$ could also reduce $IdentBias(w^{\dagger}(n); \mathcal{G}_n)$. However, there are many cases in which different IWs have the same $\epsilon_{opt}(w^{\dagger}(n))$ but significantly different $IdentBias(w^{\dagger}(n); \mathcal{G}_n)$. Furthermore, reducing $\epsilon_{opt}(w^{\dagger}(n))$ increases $IdentBias(w^{\dagger}(n); \mathcal{G}_n)$ for some cases. This observation explains small performance gains by the nested optimization problem as seen in the ablation study (cf. Appendix E.5).

