# OpenReview forum: "IW-GAE: Importance weighted group accuracy estimation for improved calibration and model selection in unsupervised domain adaptation"
_ICLR.cc/2024/Conference — Submitted to ICLR 2024_

### Official Review · Reviewer_8U9C · 2023-10-16

**Soundness:** 2 fair
**Presentation:** 2 fair
**Contribution:** 2 fair
**Rating:** 3
**Confidence:** 3

**Summary:**

The paper addresses the problem of calibration in unsupervised domain adaptation.
 The proposed method is built on the importance sampling principle. The data is split into groups and an importance weight is defined for each group. The optimal scaling temperature is defined as the one that minimizes the difference between two source domain accuracy estimations.

**Strengths:**

The paper address an important problem in which there isn't still a satisfied solution.
I think the research direction proposed by the paper has a merit.  However, the paper needs to be better written and should include a clear motivation and justification for the proposed method.

**Weaknesses:**

The paper is not clearly written. An algorithm box that states the steps of the proposed algorithm can improve a lot the paper readability.
The groups are defined in section 4.2 are exactly the groups which are defined by the ECE measure based on the confidence values.  You can use this to simplify the presentation.
In eq 8 (and eq 34) it is not clear how the optimized function depends on the temperature t?  As far as I understand the only thing  that is modified by t is the group arrangement of the target data.
My my concern is the justification of the proposed method. The algorithm looks for a temperature  that  minimizes the difference between two source domain accuracy estimations (one direct Monte-Carlo based estimation (MC) and one using importance sampling (IW)). It is not clear to me why a temperature that minimizes this accuracy estimation difference, is the one that yields improved calibration?

**Questions:**

hy a temperature that minimizes this accuracy estimation difference, is the one that yields improved calibration?

In eq 8 (and eq 34) it is not clear how the optimized function depends on the temperature t?

---

> ### Author Response · Authors · 2023-11-22
>
> (Comment) “The paper should include a clear motivation and justification for the proposed method.”
>
> (Response) For motivation, please see the “common response for the motivation for estimating group accuracy” which explains that a group accuracy estimator is statistically more favorable because the maximum likelihood estimation (MLE) of group accuracy has a lower expected mean-squared error than the MLE of individual accuracy.
>
> For the theoretical and empirical justifications of IW-GAE, kindly see the response to the last question.
>
> (Comment) “The paper is not clearly written. An algorithm box that states the steps of the proposed algorithm can improve a lot the paper readability.”
>
> (Response) We appreciate the valuable suggestion by Reviewer 8U9C. As Reviewer 8U9C recommended, we have added the pseudocode of the proposed method in Appendix C.3, which includes the full details required to implement the proposed method. We believe this addition will help readers to understand the proposed method in detail.
>
> (Comment) “The groups are defined in section 4.2 are exactly the groups which are defined by the ECE measure based on the confidence values. You can use this to simplify the presentation.”
>
> (Response) Thanks for the great suggestion. In the original submission, we aimed to present the theoretical support of using the existing group definition in the ECE measure. However, following Reviewer 8U9C’s comment, we have significantly simplified the definition of groups in Section 4.2 and now present the theoretical support in Appendix A.4.
>
> (Comment) “In eq 8 (and eq 34) it is not clear how the optimized function depends on the temperature t? As far as I understand the only thing that is modified by t is the group arrangement of the target data. My concern is the justification of the proposed method. The algorithm looks for a temperature that minimizes the difference between two source domain accuracy estimations (one direct Monte-Carlo based estimation (MC) and one using importance sampling (IW)). It is not clear to me why a temperature that minimizes this accuracy estimation difference, is the one that yields improved calibration?“
>
> (Response) Reviewer 8U9C’s confusion about the justification of the proposed method is noted. First, we want to clarify that the proposed method obtains both importance weights and the temperature parameter by solving the optimization problem. As the Reviewer 8U9C commented, the temperature parameter influences only a group construction, which can be thought of as setting the lower bound of an objective value. Given the group construction under a value of the temperature parameter, optimizing the importance weights plays a role in minimizing the objective function by finding the importance weights that minimize the difference between MC-based and IW-based estimators.
>
> Next, note that for representing prediction confidence at a sample x, IW-GAE uses the estimated accuracy of the group that contains x. Therefore, minimizing the target group accuracy estimation error is equivalent to improving the calibration error. Given this equivalence, the remaining question is how is the “difference between two source domain accuracy estimations (one direct Monte-Carlo based estimation (MC) and one using importance sampling (IW))” connected to minimizing the target group accuracy estimation error. Indeed, this connection is the core theoretical contribution of our paper, which is justified by the following two parts:
> As we explained in Equation 5, for any IW, the target group accuracy estimation error of an estimator with the IW is upper bounded by its group accuracy estimation error in the source domain multiplied by two factors.
> Proposition 4.2 shows that the source group accuracy estimation error has an upper bound that contains an objective value, i.e., the difference between MC-based and IW-based estimators.
>
> Combining these two parts, we show that minimizing the “difference between two source domain accuracy estimations (one direct Monte-Carlo based estimation (MC) and one using importance sampling (IW))” improves the calibration error. For empirical evidence that supports our main theoretical results for connecting the optimization error in IW-GAE and the group accuracy estimation error in the target domain, please see the “common response for empirical evidence supporting theoretical claims.”

---

### Official Review · Reviewer_WXHo · 2023-10-17

**Soundness:** 3 good
**Presentation:** 2 fair
**Contribution:** 3 good
**Rating:** 5
**Confidence:** 2

**Summary:**

This paper proposes a method named IW-GAE that uses estimated importance weight to measure the group accuracy, and use the measured group accuracy in the task of model calibration and the task of model selection in UDA setting.

**Strengths:**

1. This paper seems to be both theoretically solid and experimentally sound.
2. This paper is well-motivated.

**Weaknesses:**

(See questions below)

**Questions:**

For the current submitted version, I vote a boardline accept score and below are my remaining concerns.

(1) This paper proposes to achieve calibrated accracy from the perspective of group accuracy. However, it seems to me that intuitively, group accuracy is more coarse-grained than the accuracy of each individual. Thus, I am curious that, will this direction of group accuracy leads to a lower upper limit than the other branches of method?

(2) Secondly, if I am not wrong, the logic of this paper's proof seems to be, (1) the proposed optimization algorithm can optimize over the upper bound of source group accuracy estimation error. (2) the source group accuracy estimation error serves as a upper bound of the target group accuracy estimation error. Thus, because of (1) and (2) one by one, the proposed optimization algorithm works. My question w.r.t. this is that, are your two upper bounds tight enough to make the thing theoretically meaningful? Taking Eq. 5 as an example, while I do not check the math very carefully, it seems that the target group accuracy estimation error is upper bounded by the source group accuracy estimation error multipled with both M and the other item in the bracket. Thus, both these two items, if large to a certain scale, while make the optimization over the upperbound sub-meaningful. I understand that the authors have discussed that they will tighten the bound by "bound the maximum and minimum values of IW". However, I think this part should be elaborated as the tightness of the bound is important from my perspective. Besides, I also appreciate if the other impacts of this "bound the maximum and minimum values of IW" operation can be discussed. For example, will this operation hurt the current theoretical flow?

(3) The last small question I have is that, the authors claim in their conclusion that, applying their model on large language model does not get improvement. They thus conclude that, "the pre-trained large-language model is less subject to the distribution shifts". Can another potential reason behind this observation be that the proposed method is poorly scalable to larger models? I hope either discussion or experiments can be made on the scalability of the proposed method.

---

> ### Author Response · Authors · 2023-11-22
> **Response (1/2)**
>
> (Comment) “This paper proposes to achieve calibrated accracy from the perspective of group accuracy. However, it seems to me that intuitively, group accuracy is more coarse-grained than the accuracy of each individual. Thus, I am curious that, will this direction of group accuracy leads to a lower upper limit than the other branches of method?”
>
> (Response) We sincerely appreciate the Reviewer WXHo for the insightful question that deals with the main motivation of our method. As the Reviewer WXHo commented, estimating group accuracy rather than individual accuracy might seem more coarse at the first glance. However, in Section 4.1 of the revised manuscript which is a summary of the result that was originally located in Appendix B.1, we showed that estimating group accuracy is more statistically favorable than estimating individual accuracy in most machine learning scenarios where only a single label is available for each input value. Specifically, the MLE of group accuracy has a lower expected mean-squared error than the MLE of individual accuracy under relatively loose conditions.
>
> Due to its significant importance for motivating our method, the revised manuscript includes the newly added Section 4.1 which summarizes the motivation for estimation group accuracy.
>
> (Comment) “if I am not wrong, the logic of this paper's proof seems to be, (1) the proposed optimization algorithm can optimize over the upper bound of source group accuracy estimation error. (2) the source group accuracy estimation error serves as a upper bound of the target group accuracy estimation error. Thus, because of (1) and (2) one by one, the proposed optimization algorithm works. My question w.r.t. this is that, are your two upper bounds tight enough to make the thing theoretically meaningful? Taking Eq. 5 as an example, while I do not check the math very carefully, it seems that the target group accuracy estimation error is upper bounded by the source group accuracy estimation error multipled with both M and the other item in the bracket. Thus, both these two items, if large to a certain scale, while make the optimization over the upperbound sub-meaningful. I understand that the authors have discussed that they will tighten the bound by "bound the maximum and minimum values of IW". However, I think this part should be elaborated as the tightness of the bound is important from my perspective.”
>
> (Response) We thank the Reviewer WXHo for the insightful question associated with a fundamental theoretical aspect of the proposed method. Yes, as the Reviewer WXHo commented, tightness of both bounds is crucial for connecting the optimization error and estimation errors of the target and source group accuracy. Due to its central importance, we have added discussions about the conditions when the bounds are tight in the corresponding locations (page 5 for the relationship (1) and page 7 for the relationship (2)).
>
> We also have summarized the results of empirical analyses of these bounds to support our main theoretical results for connecting the optimization error in IW-GAE and the group accuracy estimation error in the target domain. Please see the “common response for empirical evidence supporting theoretical claims” for more details about the empirical evaluation.
>
>
> (Comment) “Besides, I also appreciate if the other impacts of this "bound the maximum and minimum values of IW" operation can be discussed. For example, will this operation hurt the current theoretical flow?”
>
> (Response) Again, we appreciate a very great question from the Review WXHo since bounding values usually introduces a gap between theory and practice. First, we apologize for our original statement "bound the maximum and minimum values of IW" which induces misunderstanding. It should be changed to "bound the maximum and minimum values of an IW estimation." Given the corrected statement, bounding IW estimation value influences only the estimator, so it does not affect any theoretical guarantee using properties of IWs in the paper (e.g. Proposition 4.2 and inequalities (5) and (14) in the revised manuscript). However, bounding an IW estimation value can reduce its search space from its confidence interval to a bounded CI, which may increase the optimization error. However, in practice, we observed that bounding helps to improve the performance as we intended.
>
> Due to its importance, we have added this discussion in the last paragraph of Section 4.2 of the revised manuscript.
>
> “””
> We note that bounding an IW estimation value only affects the estimator, which does not affect any theoretical guarantee based on properties of true IWs such as Proposition 4.2 and inequalities in (5) and (14). However, we remark that bounding an IW estimation may increase the optimization error due to reduced search space, although we observe that it works effectively in practice.
> “””

---

> ### Author Response · Authors · 2023-11-22
> **Response (2/2)**
>
> (Comment) “The last small question I have is that, the authors claim in their conclusion that, applying their model on large language model does not get improvement. They thus conclude that, "the pre-trained large-language model is less subject to the distribution shifts". Can another potential reason behind this observation be that the proposed method is poorly scalable to larger models? I hope either discussion or experiments can be made on the scalability of the proposed method.”
>
> (Response) Kindly see the “common response for experimental results with large-language models” which explains not only IW-GAE but also all other IW-based methods (IWCV, DEV, TransCal, CPCS) fail to improve a vanilla method and the implications of the result for future research.

---

> > ### Comment · Reviewer_WXHo · 2023-11-23
> >
> > Thanks for the author's reply.
> >
> > I have read both the replies to me and the replies to the other reviewers. W.r.t. the replies to me, I believe that they have solved my concerns at least at a "pass level". W.r.t. the discussion between the authors and the other reviewers, I think the main concern (or most crucial concern for me) now lies in the novelty (especially the novelty of the theoretical guarantee part). I have to admit that I am not really an expert on importance weighting. Thus, I will keep my rate for now and wait for the other reviewers on their following comments on the novelty (of the theory part).

---

### Official Review · Reviewer_4Yn5 · 2023-10-29

**Soundness:** 3 good
**Presentation:** 3 good
**Contribution:** 2 fair
**Rating:** 5
**Confidence:** 3

**Summary:**

This paper concentrates on addressing model calibration and model selection issues within an Unsupervised Domain Adaptation (UDA) context, where a domain shift between source and target data is present. The authors have developed an importance-weighted group accuracy estimator to effectively handle distribution shifts. Numerous experiments on UDA benchmarks demonstrate the method's superior performance in both model calibration and selection tasks, compared to other baseline methods.

**Strengths:**

1. The paper concentrates on the critical and challenging aspects of model calibration and selection issues within the realm of unsupervised domain adaptation.
2. Unlike other model selection methodologies, this paper employs an optimization problem to determine the importance weight estimation from its Clopper-Pearson, aiming for precise group accuracy estimation.
3. The estimation of group accuracy in the distribution-shifted domain is supported by a solid theoretical analysis.

**Weaknesses:**

1. The legend and meaning in Figure 1 are unclear and difficult to comprehend. The annotation for Figure 1 is excessively lengthy and could benefit from moving some sections to the main body of the paper as illustrative examples.
2. The experiments conducted in the main paper don't sufficiently verify the proposed method's effectiveness. Other datasets, such as DomainNet, and other baselines in Unsupervised Domain Adaptation (UDA), should be incorporated.
3. Several analyses and discussions residing in the appendix are crucial to demonstrating your method's effectiveness. It is advised to relocate these sections to the main body of the paper.
4. The paper asserts that the proposed method is not applicable to fixed large-language models in IW-GAE. Thus, a more comprehensive discussion and additional experiments are recommended.

**Questions:**

Please refer to the weakness part.

---

> ### Author Response · Authors · 2023-11-22
>
> (Comment) “The legend and meaning in Figure 1 are unclear and difficult to comprehend. The annotation for Figure 1 is excessively lengthy and could benefit from moving some sections to the main body of the paper as illustrative examples.”
>
> (Response) Thanks for the great suggestion. As per Reviewer 4Yn5’s comment, we have significantly improved the caption of Figure 1 by delivering the core ideas of encouraging consistency between two different estimators and an ideal case for group accuracy estimation without unnecessary details. We also have moved some descriptions to the main body of the revised manuscript for better illustration (kindly see page 2 of the revised manuscript). Specifically, we have changed the caption of Figure 1 as follows.
>
> “””
> In Figure 1(a), a shaded area for the IW-based estimator represents possible estimations from IWs in the confidence interval.
> Figure 1(b) illustrates both ideal and failure cases of IW-GAE with nine data points (red diamonds) from three groups (gray boxes). Group 1 is desirable for model calibration where the group accuracy estimation (a blue rectangle) well represents the individual expected accuracies of samples in the group. Conversely, group accuracy estimation could inaccurately represent the individual accuracies in the group due to a high variance of accuracies within the group (group 2) and a high bias of the estimator (group 3).
> “””
>
>
> (Comment) “The experiments conducted in the main paper don't sufficiently verify the proposed method's effectiveness. Other datasets, such as DomainNet, and other baselines in Unsupervised Domain Adaptation (UDA), should be incorporated.”
>
> (Response) We have performed two additional experiments with another large-scale dataset (DomainNet) and an additional UDA base model (maximum classifier discrepancy). IW-GAE consistently achieves the best performance as in other benchmark experiments. For more details, kindly see the “common response for the extensiveness of experiments.”
>
> (Comment) “Several analyses and discussions residing in the appendix are crucial to demonstrating your method's effectiveness. It is advised to relocate these sections to the main body of the paper.”
>
> (Response) We sincerely appreciate the Reviewer 4Yn5 for the valuable suggestion that helps us to significantly improve the writing of the manuscript. Following the suggestion, we have summarized the motivation for estimating group accuracy that was originally discussed in Appendix B.1 to the main body. In addition, we also summarized analyses of correlations between the optimization error, the source group accuracy estimation error, and the target group accuracy estimation error discussed in Appendix E.7 of the original manuscript in the main body. We believe the insightful feedback about the layout of the manuscript from the Reviewer 4Yn5 significantly improves the revised manuscript.
>
> For more details about the changes we made, please see the “common response for the motivation for estimating group accuracy” and “common response for empirical evidence supporting theoretical claims.”
>
> (Comment) “The paper asserts that the proposed method is not applicable to fixed large-language models in IW-GAE. Thus, a more comprehensive discussion and additional experiments are recommended.”
>
> (Response) Kindly see the “common response for experimental results with large-language models” which explains not only IW-GAE but also all other IW-based methods (IWCV, DEV, TransCal, CPCS) fail to improve a vanilla method and the implications of the result for future research.

---

### Official Review · Reviewer_Yb2X · 2023-10-29

**Soundness:** 2 fair
**Presentation:** 2 fair
**Contribution:** 2 fair
**Rating:** 5
**Confidence:** 4

**Summary:**

In this paper, the authors investigate the importance weighting technique for simultaneously addressing model calibration and model selection in the context of unsupervised domain adaptation (UDA). The authors propose a novel importance-weighted group accuracy estimator, in which the importance weight is determined through a novel optimization problem. The effectiveness of this method is validated through theoretical analysis and experiments with one UDA method on one UDA dataset.

**Strengths:**

**(+)** The problems of model calibration and model selection, addressed in this paper, hold great significance for transfer learning applications.

**(+)** The IW-GAE method relies on importance weighting, a sound technique extensively validated by prior research on model calibration and model selection.

**(+)** A thorough analysis appears to provide theoretical support for the effectiveness of the proposed method IW-GAE.

**Weaknesses:**

**(-)** The contribution's novelty is limited. The paper employs importance weighting to simultaneously address both the model calibration and selection problems. However, similar works, such as (You et al., 2019) for model selection and (Wang et al., 2020; Park et al., 2020) for model calibration, have been previously published. The primary difference between this paper and previous works appears to be the introduction of bin-wise importance weighting, as proposed in (Park et al., 2022). In summary, this paper mainly replaces the use of importance weighting in prior works with a more recent advanced importance weighting technique.

**(-)** The empirical evaluation is inadequate, rendering the conclusion less reliable. The paper only examines one UDA method, MDD, on a single UDA dataset, Office-Home, for assessing the proposed method, IW-GAE. This limited scope does not provide sufficient empirical evidence for the effectiveness of IW-GAE. It is recommended to expand the experiments to encompass more UDA methods and datasets, following relevant works (Wang et al., 2020; You et al., 2019). In conclusion, the paper lacks the necessary empirical support for IW-GAE.

**(-)** The proposed method may not be effective and practical, significantly diminishing the paper's contribution. While the empirical evaluation in the main text demonstrates the effectiveness of IW-GAE with MDD, the evaluation in the appendix reveals that IW-GAE does not outperform TransCal (Wang et al., 2020) with CDAN. Moreover, IW-GAE is considerably more complex, as it involves multiple hyperparameters compared to the hyperparameter-free TransCal (Wang et al., 2020) and DEV (You et al., 2019). IW-GAE requires at least three sensitive hyperparameters to be configured: temperature, the number of accuracy groups, and the number of bins, which does not guarantee its suitability for the model selection problem. In conclusion, the current, albeit insufficient experiments suggest that IW-GAE is not competitive and practical when compared with existing solutions.

**(-)** The paper's presentation lacks clarity due to weak organization and missing information, particularly when compared to the well-structured writing in two highly relevant importance-weighting works (Wang et al., 2020; You et al., 2019) in the context of UDA. The current submission includes excessive background and proofs while omitting essential details about the algorithm and empirical evaluations. Furthermore, the motivation and novelty of this paper in comparison to existing importance-weighting works on model calibration and model selection remain unclear.

**Questions:**

Kindly see the weaknesses for specific questions and suggestions.

---

> ### Author Response · Authors · 2023-11-22
> **Response (1/3)**
>
> (Comment) “The contribution's novelty is limited. The paper employs importance weighting to simultaneously address both the model calibration and selection problems. However, similar works, such as (You et al., 2019) for model selection and (Wang et al., 2020; Park et al., 2020) for model calibration, have been previously published.”
>
> (Response) We appreciate the Reviewer Yb2X for pointing out the unclear delivery of the novelty of the paper. As a fundamental method in statistics and simulation, importance weighting has been used to derive unbiased estimators under distribution shifts in various contexts of machine learning such as reinforcement learning, causal inference, and domain adaptation. In the context of model calibration and model selection in UDA, importance weighting is used to estimate various quantities of interest in the target domain such as calibration measures (Wang et al., 2020; Park et al., 2020) and model accuracy (Sugiyama et al., 2007; You et al., 2019). However, different from the previous research, we define a new measure of group accuracy and provide a concrete algorithm for accurately estimating the newly defined measure of group accuracy with thorough theoretical support.
>
> In addition to the novelty of defining a new measure, our approach can solve important tasks of model calibration and model selection at the same time with attractive properties.
> For model calibration, since our group accuracy estimator error is directly connected to the calibration error and its bound is theoretically shown, the proposed method can guarantee a bounded calibration error in the target domain. A bounded calibration error has not been guaranteed in the previous literature, which is a significant theoretical contribution.
> For model selection, we agree that the proposed method may seem similar to the previous work (Sugiyama et al., 2007, You et al., 2019) in the sense that all methods aim to predict the model accuracy in the target domain. However, our approach has an additional regularization factor that aims to accurately estimate the accuracy of each group, which gives more fine-grained estimation ability. The effectiveness of this additional regularization is well-supported by empirical results that IW-GAE achieves state-of-the-art performances in hyperparameter selection and checkpoint selection which are important model selection tasks.
>
> To emphasize the novelty of IW-GAE compared to existing approaches, we have added the following discussion in Section 4 of the revised manuscript.
>
>
> “””
> Once we obtain an IW estimation and a group assignment with methods described in Section 4.1, IW-GAE can estimate the group accuracy that can be used to simultaneously solve model calibration and model selection tasks with attractive properties. Specifically, for model calibration, previous approaches (Park et al., 2020; Wang et al., 2021) depend on a temperature scaling method (Guo et al., 2017) that does not provide theoretical guarantees about the calibration error. In contrast, IW-GAE uses the group accuracy estimation as an estimate of confidence. Therefore, due to the guarantees about the group accuracy estimation error (cf. Proposition 4.2 and (5)), IW-GAE enjoys a bounded calibration error. For model selection, IW-GAE uses average group accuracy computed with the dataset in the target domain as a model selection criterion. While the previous approaches (Sugiyama et al., 2007; You et al., 2020) also aim to estimate the model accuracy in the target domain, IW-GAE considers an additional regularization encouraging accurate group accuracy estimation for each group.
> “””

---

> ### Author Response · Authors · 2023-11-22
> **Response (2/3)**
>
> (Comment) “The primary difference between this paper and previous works appears to be the introduction of bin-wise importance weighting, as proposed in (Park et al., 2022). In summary, this paper mainly replaces the use of importance weighting in prior works with a more recent advanced importance weighting technique.”
>
> (Response) We want to emphasize that the simple replacement by a recent binned IW estimation technique (Park et al., 2022) would be “IW-mid” which selects the middle point in the CIs as an IW estimation. However, we develop an optimization algorithm that selects the best IW from the confidence interval, which theoretically guarantees the bounded calibration error. Its usefulness compared to simple replacement has also been shown with six different benchmarks in model calibration and selection tasks. We also want to remark that the qualitative evaluation in Appendix E.4 shows IW-GAE’s more accurate group accuracy estimation ability compared to IW-Mid (cf. Figure A.3). In order to emphasize the difference between the simple replacement and our approach, we have changed the description of “IW-Mid'' from “which selects the middle point in the CI as IW estimation and originates herein” to “which selects the middle point in the CI as IW estimation and originates herein as a simple replacement of a classic IW estimation technique (Bickel et al., 2007) by a recently proposed CI estimator (Park et al., 2022)” in Section 5 of the revised manuscript.
>
> In the revised manuscript, to more thoroughly analyze the dependency of IW-GAE on the CI estimation technique developed by Park et al. (2022), we also have applied IW-GAE to a naive confidence interval by setting maximum and minimum values of IWs as the CIs. Even without a sophisticated CI estimation by Park et al. (2022), IW-GAE outperforms all baseline models by a large margin (at least 24% lower ECE on average). We believe that the result of this new ablation study shows a strong potential of IW-GAE as a general method to select IWs from CIs. To emphasize the applicability of IW-GAE with other CI estimators, we have added the following paragraph in Appendix B.2 of the revised manuscript.
>
> “””
> Our concept of determining the IW from its CI can be applied to any other valid CI estimators. For example, by analyzing a CI of the odds ratio of the logistic regression used as a domain classifier (Bickel et al., 2007; Park et al., 2020; Salvador et al., 2021), a CI of the IW can be obtained. As an extreme example, we apply IW-GAE by setting minimum and maximum values of IWs as CIs in an ablation study (Table A5). While IW-GAE outperforms strong baseline methods (CPCS and TransCal) even under this naive CI estimation, we observe that its performance is reduced compared to the setting with a sophisticated CI estimation discussed in Section 3. In this regard, advancements in IW estimation or CI estimation would be beneficial for accurately estimating the group accuracy, thereby model selection and uncertainty estimation. Therefore, we leave combining IW-GAE with advanced IW estimation techniques as an important future direction of research.
> “””
>
>
>
> (Comment) “The empirical evaluation is inadequate, rendering the conclusion less reliable. The paper only examines one UDA method, MDD, on a single UDA dataset, Office-Home, for assessing the proposed method, IW-GAE. This limited scope does not provide sufficient empirical evidence for the effectiveness of IW-GAE. It is recommended to expand the experiments to encompass more UDA methods and datasets, following relevant works (Wang et al., 2020; You et al., 2019). In conclusion, the paper lacks the necessary empirical support for IW-GAE.”
>
> (Response) We sympathize now for confusing Reviewer Yb2X about the comprehensiveness of experiments in the manuscript. In the Appendix of the original manuscript, we included additional experiments with a different UDA baseline (CDAN) and a different dataset (VisDa 2017). Also, we performed another important model selection task of choosing the best checkpoint during training. In these additional experiments, IW-GAE consistently achieves the best performances among important baselines we consider in the paper (Guo et al., 2017; Park et al., 2020; Wang et al., 2020; Sugiyama et al., 2007; You et al., 2019).
>
> In the revised manuscript, as per Reviewer Yb2X’s valuable suggestion, we perform two additional benchmark experiments with a different large-scale dataset (DomainNet) and a different base UDA method (maximum classifier discrepancy) in order to match the comprehensiveness of our empirical verification with previous works (You et al., 2019; Wang et al., 2020; Park et al., 2020). In the new benchmarks, IW-GAE consistently achieves the best performances. For more details, kindly see the common response “common response for the extensiveness of experiments.”

---

> ### Author Response · Authors · 2023-11-22
> **Response (3/3)**
>
> (Comment) “the evaluation in the appendix reveals that IW-GAE does not outperform TransCal (Wang et al., 2020) with CDAN.”
>
> (Response) We kindly refer Reviewer Yb2X to Appendix E.2 for additional experiments with a different base model (CDAN) where IW-GAE outperforms TrasCal (Wang et al., 2020) by 2%, which was given in the original manuscript. In addition, in our new experiments that we performed thanks to the suggestion by Reviewer Yb2X, IW-GAE consistently outperforms TransCal in different benchmarks with a different dataset (DomainNet) and a different base model (MCD).
>
> (Comment) “IW-GAE is considerably more complex, as it involves multiple hyperparameters compared to the hyperparameter-free TransCal (Wang et al., 2020) and DEV (You et al., 2019). IW-GAE requires at least three sensitive hyperparameters to be configured: temperature, the number of accuracy groups, and the number of bins, which does not guarantee its suitability for the model selection problem. ”
>
> (Response) We now agree with Reviewer Yb2X’s confusion about hyperparameters included in IW-GAE and not effectively referring to relevant analyses that we performed in the original submission. First of all, we want to note that hyperparameters involved in IW-GAE are only the number of groups and the number of bins, not the temperature parameter. Specifically, the temperature parameter is optimized within the search space in IW-GAE, which is the same as with the previous methods (Wang et al., 2020, You et al., 2019). Specifically, in the literature (Wang et al., 2020, You et al., 2019), the search space is defined as a box constraint, i.e., setting the maximum and minimum values of the temperature parameter. However, in IW-GAE, it is defined as a discrete space due to the non-smoothness of the optimization problem. From this perspective, we agree that IW-GAE involves a more complex optimization procedure compared to previous literature. However, we believe that achieving state-of-the-art performances in both model calibration and selection tasks under three popular UDA methods (MDD, CDAN, and MCD) on three different datasets (OfficeHome, VisDa-2017, and DomainNet) can account for the more complex optimization procedure involved in IW-GAE.
>
> Also, we performed the sensitivity analysis with respect to all hyperparameters involved in IW-GAE (the number of groups and the number of bins) in Appendix E.6, which shows the robustness of performances of IW-GAE under changes in these hyperparameters. Also, we want to emphasize that we perform all experiments with three different architectures on three different datasets with a single fixed value of the hyperparameter, which again shows the robustness of IW-GAE for the hyperparameter choice. As it might not be easy to detect the citation of additional experiments that include sensitivity analysis in the original manuscript, we have added a new section titled “5.3. Qualitative evaluation, ablation study, and sensitivity analysis.”
>
> (Comment) “The paper's presentation lacks clarity due to weak organization and missing information, particularly when compared to the well-structured writing in two highly relevant importance-weighting works (Wang et al., 2020; You et al., 2019) in the context of UDA. The current submission includes excessive background and proofs while omitting essential details about the algorithm and empirical evaluations. Furthermore, the motivation and novelty of this paper in comparison to existing importance-weighting works on model calibration and model selection remain unclear.”
>
> (Response) We appreciate the Reviewer Yb2X for pointing out the weak organization that includes the important details in the Appendix and does not include the essential details about the algorithm and evaluations. Following the Reviewer Yb2X’s valuable feedback, we have added the pseudocode of the proposed method to provide full details of the algorithm required to implement the proposed method in Appendix C.3 based on clearly written pseudocode of the previous work (Wang et al., 2020). In addition, we also have added pseudocodes for evaluating methods for model calibration and model selection in Appendix C.3. Please, kindly see Algorithms 1, 2, and 3 on pages 17-18 of the revised manuscript. We believe these additions will greatly help readers to understand the proposed method in detail.
>
> Also, thanks to valuable feedback from the Reviewer Yb2X, we have significantly simplified the explanation of the previous method (Park et al., 2022) by excluding unnecessary details and derivations. For the change, kindly check pages 3-4 of the revised manuscript.
>
> Finally, kindly see the first response for the novelty of the paper compared to relevant works and the “common response for the motivation for estimating group accuracy”  for the motivation for estimating group accuracy.

---

> > ### Comment · Reviewer_Yb2X · 2023-11-23
> > **Thank you for the response**
> >
> > I appreciate the authors' efforts in responding to my queries and addressing some concerns related to clarity. However, significant concerns persist regarding the effectiveness of the proposed method in practical unsupervised domain adaptation scenarios. I maintain my stance against the repeated claim of "achieving state-of-the-art performances in both model calibration and selection tasks." As I initially pointed out, it is advisable to broaden the experimental scope by including more UDA methods and datasets, at least aligning with relevant works (Wang et al., 2020; You et al., 2019), respectively
> >
> > Considering the authors' dedicated revision and responses, I have decided to increase my score to a borderline rejection of "5". Nevertheless, I still believe that the paper requires additional empirical evidence to substantiate the bold assertion of addressing two challenging problems simultaneously.

---

> > > ### Author Response · Authors · 2023-11-23
> > >
> > > We appreciate so much the Reviewer Yb2X for acknowledging our responses to the comments about our contribution's novelty, motivation for our approach of estimating group accuracy, and clarity of the proposed algorithm, except for the extensiveness of our experiments. However, we want to claim that the extensiveness of our empirical verification is at least comparable with relevant works (You et al., 2019; Wang et al., 2020; Park et al., 2020).
> > >
> > > Specifically, compared to You et al (2019), we did not perform experiments on a toy synthetic dataset because the dataset is for the regression task and the Digits dataset because the dataset is for generative models, which are out of the scope of our paper. Also, we note that the Office-31 dataset, which is tested in You et al (2019) but not in our paper, is much simpler and easier to deal with compared to the Office-Home dataset, which is tested in our paper but not in You et al (2019), which has more samples, more classes, and more domains.
> > >
> > > Also, compared to Wang et al (2020), we match the extensiveness of our experiments in terms of datasets (Office-Home, VisDA, and DomainNet), except small-scaled Office-31 and ImageNet-Sketch which has similar characteristics with VisDA considering both contain real-world images and synthetic images. Also, our experiments match the extensiveness of popular UDA base models (CDAN, MDD, and MCD), except partially tested models (AFN, BNM, DAN, and JAN) in Wang et al (2020) with only one dataset. We also want to remark that Wang et al (2020) included the results from the Multi-Domain Sentiment dataset, but we could not find the corresponding code from the official repository, and the details about the base model, preprocessing of the data, and the training configuration are missing in the paper, preventing us to reproduce the results.

---

### Author Response · Authors · 2023-11-22
**Common Responses (1/2)**

We sincerely appreciate the reviewers for their time and efforts to give insightful feedback. Also, we are happy to see that the majority of reviewers appreciate the value of our approach supported by “a solid theoretical analysis” that deals with “the critical and challenging aspects of model calibration and selection issues” which “hold great significance for transfer learning applications.” It is important to note and stress that the majority of comments are about discussions and analyses that we place in the Appendix of the manuscript, which are now summarized in the main body. Thanks to the reviewers’ valuable comments, we have significantly improved the manuscript and uploaded the revised manuscript that highlights the changes in blue color. Below, we have summarized common comments and major changes we have made from the reviewers’ valuable comments.


# Common response for the motivation for estimating group accuracy
In Appendix B.1 of the original manuscript, we showed that estimating group accuracy is statistically more favorable than estimating individual accuracy when only a single label is available for each distinct input value which is the case for most machine learning benchmark datasets. Specifically, we prove that the maximum likelihood estimator (MLE) of group accuracy has a lower expected mean-squared error than the MLE of individual accuracy under relatively loose conditions.

Due to its significant importance for motivating our method, the revised manuscript involves the newly added Section 4.1 which summarizes the motivation for estimation group accuracy.


# Common response for empirical evidence supporting theoretical claims
In Appendix E.7 of the original manuscript, we presented the empirical verification of theoretical guarantees in our paper. Specifically, we gathered IWs under all temperature parameters in all subtasks in the OfficeHome dataset, which involves 720 IWs. Then, we analyze the correlations between (1) the target and source group accuracy estimation errors, (2) the source group accuracy estimation error and the optimization error, and more importantly, (3) the target group accuracy estimation error and the optimization error.

In the analyses, we observed meaningful correlations in all three cases, albeit the correlation is somewhat lower for case (3). Specifically, IWs with a low source group accuracy estimation error tend to have a low target group accuracy estimation error. Also, when the optimization error decreases, both source and target group accuracy estimation errors tend to decrease. These results validate our approach of minimizing the optimization error for reducing the target group accuracy estimation error, supporting our main theoretical claims.

Considering their significant importance, we have summarized the analyses in the main body of the revised manuscript (pages 5 and 7) with new figures in Figure 2.


# Common response for the extensiveness of experiments
We performed two additional experiments with a different UDA base model (maximum classifier discrepancy; MCD) and a different large-scale dataset (DomainNet). Consistent with other experiments in the paper, IW-GAE achieves the best performances among all baselines, outperforming the second-best method by 31% and 5% in terms of the average ECE. For more details, kindly see Table A.1 and Table A.3 in the Appendix of the revised manuscript.

Thanks to reviewers’ valuable suggestion, we believe that the extensiveness of our empirical evaluation, which includes verifications with the three most popular base models (CDAN, MDD, and MCD) on three standard datasets (OfficeHome, VisDa-2017, and DomainNet), is on par with previous literature (You et al., 2019; Wang et al., 2020; Park et al., 2020), except for the exclusion of small-scale datasets such as the synthetic toy regression data, the Office-31 dataset, and the Digits dataset.

---

> ### Author Response · Authors · 2023-11-22
> **Common Responses (1/2)**
>
> # Common response for experimental results with large language models
> We were unaware of the confusion related to the results of the large-language model experiment. Actually, all IW-based methods discussed in the paper (IWCV, DEV, TransCal, CPCS, IW-GAE) fail to improve standard methods in the i.i.d. scenario (vanilla and temperature scaling for model calibration; cross-validation for model selection). Therefore, the result does not indicate the limited scalability of IW-GAE. In order to prevent confusion related to failures of IW-based methods on large-language models and remark on the important research questions, we have modified the paragraph regarding the large-language models in the conclusion as follows.
>
> """
> Finally, we note that all IW-based methods (CPCS, IW-TS, TransCal, IW-GAE) fail to improve the standard method in the i.i.d. scenario in our experiments with pre-trained large-language models (XLM-R (Conneau et al., 2019) and GPT-2 (Solaiman et al., 2019)). We conjecture that these models are less subject to the distribution shifts due to massive amounts of training data that may include the target domain datasets, so applying the methods in the i.i.d. setting can work effectively. In this regard, we leave the following important research questions: “Are IW-based methods less effective, or even detrimental, under mild distribution shifts?” and “Can we develop methods that work well under all levels of distribution shifts?”
> """

---

### Meta-Review · Area_Chair_oV4Y · 2023-12-06

**Metareview:**

This paper investigates the group accuracy estimation problem under data distribution shift using an importance weighting technique. The resulting method can benefit model calibration and model selection in the context of unsupervised domain adaptation (UDA). The main innovation is a novel optimization problem for importance weights estimation with theoretical guarantees. The effectiveness of this method is validated in experiments with UDA datasets.

Strength: the reviewers all agree that this problem is important and the proposed method is sound.

Weakness: the major concerns that remain after the rebuttal are the novelty and the presentation. For novelty, the main idea is to utilize the bin-wise important weights, which is a relaxed reformulation of a previous method. For presentation, multiple reviewers mention it is better to organize the content so that the contribution and the difference compared to previous methods are more clear.

**Justification For Why Not Higher Score:**

This paper is slightly below the acceptance line, both the scores from the reviewers and the discussion reflects that.

**Justification For Why Not Lower Score:**

N/A

---

### Decision · Program_Chairs · 2024-01-16

Reject